# How do medical students deal with the topic of racism? A qualitative analysis of group discussions in Germany

**Simon Matteo Gerhards***, **Mark Schweda**

Division for Ethics in Medicine, Department for Health Services Research, Carl von Ossietzky University of Oldenburg, Oldenburg, Germany

* simon.matteo.gerhards@uni-oldenburg.de

## Abstract

### Background

Anti-racism is part of the medical professional ethos. Nevertheless, racism pervades medicine on individual, institutional, and structural levels. The concept of habitus helps to understand deficiencies in enacting anti-racism in practice. We use a habitus-based framework to analyse how medical students in Germany deal with the topic of racism. The research questions are: What are medical students' understandings of racism? How do they deal with the topic in discussions? What difficulties do they face in such discussions?

### Methods

In a qualitative-explorative research design, we conducted six online group discussions with 32 medical students from medical schools all over Germany. Data analysis combined qualitative methods from thematic qualitative content analysis and the documentary method.

### Results

We identified five typical ways of dealing with the topic of racism in discussions. The first one ('scientistic') orientates action towards the idea of medicine as an objective science, justifies the use of racial categories as scientific, and defines racism based on intention. The second ('pragmatic') orientates action towards tacit rules of clinical practice, justifies the use of racialised categories as practical and defines racism as an interpersonal problem. The third ('subjectivist') lacks a clear orientation of action for dealing with the topic of racism and instead displays uncertainty and subjectivism in understanding racialised categorisations as well as racism. The fourth ('interculturalist') orientates action towards an ideal of intercultural exchange, understands racialised categorisations as representing cultural differences and interprets racism as prejudice against cultures. The fifth ('critical') orientates action towards sociological scholarship, understands racialised categorisations as social constructs and views racism as a structural problem.

**Data Availability Statement:** De-identified data cannot be shared due to ethical and legal restrictions connected to the qualitative nature of the data. The data-set contains potentially

identifying and sensitive participant information. Therefore, the Institutional Review Board of the School for Medicine and Health Sciences, University of Oldenburg (No. 2021-080) imposed a restriction on making the data available. Contact: Medical Ethics Committee, School VI – Medicine and Health Science, University of Oldenburg, Ammerlaender Heerstraße 114-118, D-26129 Oldenburg. E-mail: med.ethikkommission@uni-oldenburg.de.

**Funding:** This study received funding from the School of Medicine and Health Sciences, University of Oldenburg in the form of the "Dr. med.-Exzellenzprogramm - Sommer Semester 2023" research grant awarded to SG.

**Competing interests:** The authors have read the journal's policy and have the following competing interests: SG is part of the student organization "Medicine for Antiracist Action," a project of the Federal Representation of Medical Students in Germany (bvmd e.V.). This does not alter our adherence to PLOS ONE policies on sharing data and materials.

## Conclusion

The results presented help to understand preconditions of enacting anti-racism in medicine and point to difficulties and learning needs. The heterogenous ways of dealing with the topic require a differentiated approach in medical education.

## Introduction

Prominent professional codices and principles declare that physicians should reject racism [1–3]. In the "Declaration of Berlin", the World Medical Association even urges all physicians to "commit to actively work to dismantle racist policies and practices in health care and advocate for anti-racist policies and practices that support equity in health care and social justice" [4].

This example emphasises the professional responsibility of physicians to act against racism in medicine and health care [5] and amounts to an expressly anti-racist professional ethos that "actively seeks to identify, remove, prevent and mitigate racially inequitable outcomes and power imbalances between groups and change the structures that sustain inequalities" [3, 6]. Nevertheless, empirical research shows that racism pervades medicine and health care on individual [7, 8], institutional, and structural levels [9–21]. This discrepancy between professional ethos and practice can be described as an insufficient enactment of professional anti-racism [22].

Medical education is central to translating professional ideals into practice. It shapes physicians' professional socialisation [23], the interactional process of becoming and being a member of an occupational group by which they acquire explicit and implicit knowledge, skills, and attitudes from the professional environment [24, 25]. It thus promotes the development of the professional medical habitus [26–28], a system of acquired dispositions of "perception, thought and action" [27] comprising tacit knowledge and attitudes that orientate medical students' actions regarding specific topics, problems, or situations [29, 30] such as racism.

Breaking the silence and enabling discussions about racism in medical education can be an important step to tackling the persistence of racial discrimination in health care [31–33]. Talking about racism is a precondition for inducing reflection, decreasing racist beliefs and discrimination, and expanding critical action [33–38]. Nevertheless, research also shows that discussing racism faces a variety of challenges such as ignorance [39, 40], denial [41], negative feelings, defensive behaviour [37, 42, 43], and may lead to the acute recurrence of racist discrimination [34, 44, 45].

So far, only few qualitative studies exist on medical students' perspectives on racism and their ways of dealing with the topic in discussions [42, 46–48]. Novak and colleagues (2022) [47] propose a model of "stances" that ranges from less critical to more critical and aims to represent the scope of students' attitudes towards anti-racist action. For the German context, Hallal (2015) describes that medical students react with "denial", "defence" and "minimalization" as well as "adaption" and "integration" in discussions about the diversity of patients [42]. She also points to the crucial and sometimes detrimental influence of professional socialisation and habitus on how medical students perceive diverse patient groups [42]. Roberts et al. (2007) found in a qualitative focus group study that medical students have difficulties discussing race and racism and are more at ease when a biomedical model frames discussions about race [48].

Against the backdrop of a general lack of research about racism and anti-racism in the context of medicine and health care in Germany [15–17, 49], we seek to explore how medical students deal with the topic of racism in discussions. Our research questions are: What are medical students' understandings of racism? How do they deal with the topic in discussions? What difficulties do they face in such discussions?

To address these questions, we conducted an explorative study comprising a qualitative analysis of six online group discussions with 32 medical students in Germany on the topic of racism in medicine and health care. In the following, we first consider the state of research on racism, race, and racial bias in the context of medicine and health care. We then provide results from our qualitative analysis of the group discussions, fleshing out five typical ways of dealing with the topic of racism in discussions: 'scientistic', 'pragmatic', 'subjectivist', 'interculturalist', and 'critical'. Considering their respective specificities, their various interactions, and their connections to the professional medical habitus, we discuss how medical education can contribute to the implementation of anti-racism into practice.

## Racism, race and racial bias in medicine and health care

Defining racism is a matter of ongoing debate and there is neither societal nor scientific consensus [50, 51]. Racism is often perceived as a morally charged term [52], not as a descriptive or analytical concept [53]. Different definitions include or exclude different phenomena and may ultimately motivate different approaches to anti-racism [50]. Definitions in the social sciences that are often referred to emphasise that racism is not simply a phenomenon of explicit or intended racist action nor an individual prejudice [54]. Instead, scholars as well as activists stress the unconscious and collective aspects of racism as an ideology and societal structure [50]. Philomena Essed, for example, defines racism as

> an ideology, a structure, and a process by which certain groups are regarded as inherently different and inferior 'races' or ethnic groups on the basis of actual or ascribed biological or cultural characteristics. Subsequently, these differences serve to explain the exclusion of members of these groups from access to material and non-material resources. [55] (own translation)

As real, biologically distinct populations, races do not exist. They attain their significance as social constructs [56–58]. The process in which racial groups are socially constructed is also called racialisation [59]. Miles sees racialisation as "a process of categorisation, a representational process of defining an Other, usually, but not exclusively, somatically" [57]. Acknowledging that there is controversy about the use and meaning of *racialisation* [60, 61], we will use the term *racialised* for groups and categories that are created and used based on an ideology of racism [55, 59] as well as for the external assignment of individuals to these groups. With this, we aim to do justice to the fact that what is understood as races or racial groups and who is affected by racism may vary with contexts and show regional differences [4, 62, 63].

In health research, racism is often but inconsistently operationalised on different levels, such as individual, institutional and structural [9–11, 64–66]. It is acknowledged as a social determinant affecting people's health through various pathways [4, 20, 66–69]. As a concept entangled with the history of colonialism and racial anthropology in Western sciences and medicine, it is also ingrained in the practice and knowledge of medical professionals [70–75]. In his seminal work "Black Skin, White Masks", Frantz Fanon gives examples of how racism structures people's habitus in terms of "implicit knowledge", and "corporeal schema[ta]" [76]. Fanon describes how dominant narratives and collective prejudices ("legends, stories, history")

promote racist objectification and dehumanization [76]. In this sense, his analysis of physicians' unconscious enactment of racism illustrates the influence of habitus. He describes how physicians may refer to their good intentions when criticised for derogatory behaviour towards black patients: "Yet, we'll be told, there is no intention to wilfully give offense" [76]. For Fanon, it is exactly this "indifference, this automatic manner of classifying" [76] that is problematic: aspects of physicians' behaviour that can be understood to stem from a specific (professional) socialisation and habitus.

More recent research on racial bias takes a closer look at cognitive processes that influence such discrepancy between intention and action. The results underline that implicit attitudes and behaviours of health care providers contribute to health disparities. Bias is the "negative evaluation of one group and its members relative to another" [77]. It can be explicit when a person is aware of their evaluation, perceives it as correct, and has the motivation to act accordingly in an environment that permits it [77]. Implicit bias is unintentional and operates on less conscious levels. Psychological research has shown that implicit bias is activated automatically and unknowingly by specific prompts such as phenotypical aspects associated with racialisation [77, 78] and has an influence on "judgments, decisions, and behaviors" [79]. A widely used method to measure implicit bias is the latency-based Implicit Association Test (IAT) [79, 80].

The "Unequal Treatment" report published in 2003 by the national Institute of Medicine in the USA stressed the need for more research to understand better how "bias, stereotyping, prejudice and clinical uncertainty on the part of health care providers may contribute to racial and ethnic disparities in health care" [81]. This research initiated by the US National Medical Association was the starting point for numerous studies investigating racial bias in medicine and health care, mainly in the US context [81–83]. The main results from this research can be summarised as follows [8, 82–85]: Even if physicians may perceive themselves as unbiased, they generally show the same levels of implicit bias as other people [84, 86]. Implicit racial bias may influence the quality of health care provision, e.g., with regard to diagnosis and treatment decisions [83, 86], and is higher in physicians working in a stressful environment with high workload [87]. Physicians' bias against non-white racialised patients may lead to doctor-patient interactions that are perceived as negative by patients [8, 88, 89]. Racial bias also has an influence on health research [90] and may therefore also be found in the knowledge base of medical practice and impede an equal provision of health care, e.g., via physicians' biased assumptions about biological race-differences [70], (clinical) algorithms [12, 73, 91], or biased technical devices [92, 93]. As the explicit and implicit attitudes of physicians often diverge, studies on bias should include implicit and unconscious aspects [84]. Although the concept of implicit bias is widely used to explain differences between health care professionals' explicit intentions and their actions, factors influencing the formation of implicit bias still pose questions [79, 94].

While recent research on racism in medicine and health care concentrates on the US context [16], there is a lack of systematic studies on the situation in Germany [95, 96]. Twenty-two percent of the German population report experiences of racism [43] and there is emerging scientific evidence for racism in the German health care system [15, 17, 49, 88, 97–100]. The existing research mainly focuses on the perspectives of patients, physicians, and other health professionals. Apart from a few exceptions [42, 49], the perspectives of medical students on racism have not yet been explicitly studied in the German context. As future doctors in an intensive phase of their professional socialization, their perspectives are crucial for understanding and tackling racism in medicine and health care.

A multitude of educational approaches have been developed to address racism in medicine and health care [45, 101–107]. Activism [108, 109], social sciences [110, 111] and pedagogy

[112] all provide foundations for approaches to anti-racism in educational contexts. Attempts have been made to integrate anti-racism into medical education curricula. Osei-Tutu et al. (2023) propose to incorporate anti-racist concepts in the CanMEDS Physician Competency Framework, which is used for curriculum design in medical education worldwide [3, 113].

In the context of medical education in Germany, the revised German National Competence Based Catalogue of Learning Objectives for Undergraduate Medical Education (NKLM 2.0) now requires medical students to attain competencies of anti-racism. The corresponding learning objective (VIII.6-04.4.13) formulates that medical students learn to "identify disadvantages, stigmatization and discrimination on racist grounds or because of ethnic origin" and "direct action towards the prevention or elimination of these disadvantages" (own translation [114]).

## Methods

In order to explore medical students' ways of dealing with the topic of racism in discussions, we conducted six semi-structured online group discussions with 32 medical students from 13 German medical schools. This approach had several advantages: Group discussions facilitate the collection of a range of different positions and arguments regarding a given topic. Moreover, they also allow for an analysis of aspects of performance and group dynamics that are not accessible through individual interviews. Eventually, due to their casual atmosphere and interactional qualities, group discussions create conditions that are comparable to how medical students would discuss the topic of racism in the context of medical education.

Since we aimed to explore how medical students deal with the topic of racism in discussions that might take place in common teaching settings in Germany, we opted for mixed groups, including both students with and without personal experiences of racism. Discussions about racism not only arise in specifically prepared teaching environments but often rather casually in everyday clinical or educational situations involving a variety of students. To understand what social mechanisms and group dynamics may develop in such discussions among heterogeneous student groups, most of the group discussions in our study involved both participants with and without personal experience of racist discrimination.

Recruitment took place from 8th June to 15th September 2021. For recruitment, we distributed a public call via student groups and offices, lecturers, and social media. Moreover, we used snowball sampling. Inclusion criteria were enrolment in medical studies at a university in Germany, age of majority, and sufficient German language skills to follow the discussion and contribute to it. We used a pre-questionnaire to collect information on socio-demographic aspects such as age, gender, year of study, discrimination experience, and political engagement. For discrimination experience and political engagement, we offered space for open-ended text responses.

In total, 41 individuals replied to the call and 40 met the inclusion criteria. Eight did not take part in the discussions, either because they did not follow our invitation or dropped out due to scheduling difficulties. Of the 32 participants, eight identified as male, 23 as female, and one person identified as diverse. The age of participants ranged from 18 to 31 years, and they were all in their first to sixth year of medical studies. Six participants reported that they had experienced discrimination due to aspects that fall under the above-mentioned definition of racism. Twelve noted that they were politically active. Due to the online setting and data saturation, the sample size was limited to six online group discussions with four to seven participants each. Table 1 shows the participants' characteristics.

The discussions were held between July and September 2021 with the video conference tool Webex. They lasted between 90 and 120 minutes and were moderated by two members of the

**Table 1. Sample characteristics.**

| Total, (n) | 32 |
|---|---|
| **Self-identified gender** | |
| Male, (n) | 8 |
| Female, (n) | 23 |
| Diverse, (n) | 1 |
| **Age [years]** | |
| Age, (range) | 18–31 |
| Age, (mean) | 24,8 |
| **Academic year** | |
| 1st year, (n) | 5 |
| 2nd year, (n) | 5 |
| 3rd year, (n) | 3 |
| 4th year, (n) | 8 |
| 5th year, (n) | 7 |
| ≥ 6th year, (n) | 3 |
| NI, (n) | 1 |
| **Experience of discrimination** | |
| Racism, (n)<br>"racialisation," "origin", "migration background," "religion", "racism"[1] | 6 |
| Other, (n) | 9 |
| No information, (n) | 17 |
| **Politically engaged participants, (n)** | 12 |

Note: [1] Buzzwords extracted from the open-ended responses to the prompt: "I experience discrimination due to. . .".
Multiple answers were possible.

research team. A third person from the team was present in the video conferences without video transmission to take field notes without participating in the discussions.

All members of the research team self-identify as white and are also perceived as white by others. SG moderated the discussions. The study is part of his doctoral thesis. As a final-year medical student at the University of Oldenburg, he knew some of the participants personally. Since 2020, he has also been an active part of the federal student project "Medicine for Anti-Racist Action (MAA)" that brings together medical students, physicians, and other health professionals all over Germany with and without personal experience of racism for activities such as organising a regular online journal club, developing training material on racism and anti-racism in health care, stakeholder advocacy and reflection groups about white supremacy for medical students [116, 117]. The other part of the moderation team was a female postdoctoral researcher at the same university. She taught medical ethics and therefore also knew a small number of participants from her classes. Both were experienced in qualitative research but conducted online group discussions for the first time. MS is a senior researcher in medical ethics and participated in some of the group discussions but stayed in the background to gather field notes without video transmission. SG and MS wrote this manuscript.

We took specific precautions in the preparation of the discussions since the specific setting can lead to the recurrence of racist discrimination. We asked the participants to interact respectfully and trustfully, and they agreed to abide by previously communicated discussion rules, such as respectful and non-offensive language. Participants were also informed in advance that despite these precautionary measures, no totally safe space could be guaranteed, and racism and micro-aggressions might occur during the group discussions. In addition to

obtaining written informed consent, we held one-on-one online meetings with all participants to convey this information and provide room for open questions and uncertainties. During the online discussions, the moderation team was always available for private communication via direct messages and was prepared to intervene if necessary. Moreover, all participants received a list of contact persons and anti-discrimination organisations in Germany. IRB approval for the study was obtained from the Medical Ethics Committee of the School for Medicine and Health Sciences, University of Oldenburg, Germany (No. 2021–080). The study was performed in line with the principles of the Declaration of Helsinki.

The semi-structured discussion guidelines included questions about personal experience with or witnessing of racism, awareness and understanding of interpersonal, institutional and structural racism, and demands for medical education (see S1 Text. Discussion guide). The discussions were audio recorded, transcribed verbatim, and then pseudonymised. We organised and analysed the data using MAXQDA [88]. For this article, the quotes from our material were translated into English by the authors and speaker names replaced with pseudonyms.

As indicated above, the qualitative data analysis was guided by the assumption that professional habitus influences peoples' dealing with racism [52, 76, 115, 116]. The habitus concept proved helpful to analyse human behaviour by linking the structural aspects of a social field with subjective agency and was previously used to analyse the processes of professional socialization in health care and medicine [26, 28]. Following this idea, our research focused not only on explicitly verbalised thoughts but aimed to study more encompassing habitus-related aspects in medical students' discussions about the topic of racism. Therefore, we analysed the group discussions on two levels: *what* the participants said about racism and *how* they discussed the topic. On the level of how medical students discuss the topic of racism, the analysis aimed to reconstruct *orientations of action*, that is, the tacit knowledge that guides peoples' action with regard to a specific topic, problem or situation without being necessarily explicitly expressed [117]. See Table 2 for an overview of key concepts that formed the analysis' conceptual framework.

To tackle the two foci of analysis, we combined two methods of qualitative analysis: a) thematic content analysis [119] and b) an adapted approach of the documentary method [30, 120]. The thematic content analysis provided us with a first overview of medical students' perspectives on the topic of racism. The documentary method aimed for the reconstruction of

**Table 2. Conceptual framework for analysis.**

| Concept | Definition |
| --- | --- |
| **Racism** | "An ideology, a structure and a process by which certain groups are regarded as inherently different and inferior 'races' or ethnic groups on the basis of actual or ascribed biological or cultural characteristics." [55] (own translation) |
| **Race** | Not a biological category but as a social category resulting from racism and racialisation. [74] |
| **Racialisation** | The process in which racial groups are socially constructed including the external assignment of a person to a racial group. [57] |
| **Professional socialization** | Interactional process of becoming and being member of an occupational group by which individuals acquire explicit and implicit knowledge, skills and attitudes from the professional environment. [24, 25, 118] |
| **Habitus** | Systems of dispositions orientating perception, thought and action that are acquired by socialization. [27, 30] |
| **Orientation of action** | Tacit knowledge and attitudes that guides peoples' action with regard to a specific topic, problem or situation without being necessarily explicitly expressed. Multiple orientations of action form the habitus. [30] |

habitus-related aspects, the orientations of action that underly medical students' ways of dealing with the topic of racism.

We first conducted a thematic content analysis [119] of *what* medical students say about racism to structure the data according to content-related aspects such as: *problem awareness, understanding of interpersonal/institutional/structural racism, understanding of racialised categorisations, definition of racism, understandings of mechanisms of racism, affectedness* and *non-affectedness*, and *speaking about racism* [97, 121]. To this end, the transcripts of the discussions were coded by two researchers following the guidelines for a thematic qualitative analysis proposed by Kuckartz [119]: First, deductive codes were defined in a codebook based on research questions and discussion guidelines. In a second round of coding, inductive thematic aspects were coded to differentiate the deductively coded material and to identify additional aspects mentioned by the participants. The whole coding process was accompanied by close exchange and discussion about the definition and application of codes in the research team.

In a second step based on the reconstructive qualitative approach of the documentary method [120], we analysed *how* medical students discussed racism by focusing more on implicit a-theoretical orientations of action shaping their ways of dealing with the topic in discussions [120]. To this end, we analysed selected passages of the group discussion transcripts with special attention to the performative and interactive modes of discussion [122]. The procedure of qualitative analysis based on the documentary method was threefold:

1. In a first step that is called the "formulating interpretation" [120], thematic aspects of selected passages were examined by reconstructing *what* was explicitly said by participants. This "formulating interpretation" is about reformulating only the explicit meaning of *what* was said without further interpretation and results in a sequence of topics of the analysed passage [29].

2. In the second step, we then analysed *how* the students discussed the topic of racism by analysing the organisation of discourse and comparing ways of discussions between different passages, participants, and group discussions [120]. This step is called the *reflective analysis* and aims to reconstruct the underlying *orientations of actions* shaping the participants' ways of discussing a topic [29, 30].

3. By comparing and contrasting the reconstructed *orientations of action*, our analysis resulted in a characterization of medical students' typical ways of dealing with the topic of racism in discussions. The overview is based on the reconstruction of *orientations of action* in their connection to medical students' a) *understandings of racialised categorisations* and b) *understandings of racism*. The empirical analysis of these levels is integrated in the synopsis of the results so that every way of dealing with the topic is characterised with regard to its *orientation of action, understanding of racialised categorisation* and *understanding of racism*.

## Results

We identified five typical ways in which medical students deal with the topic of racism in discussions: 'scientistic', 'pragmatic', 'subjectivist', 'interculturalist', and 'critical' (for a synopsis, see Table 3). These terms are used to contour the respective sets of characteristics inductively reconstructed from the empirical material. They are not to be confused with scientific concepts deductively applied to the material. Moreover, they do not stand for individual participants but for a range of ways of dealing with the topic of racism in discussions. Hence, one person can display different characteristics.

**Table 3. Synopsis of medical students' typical ways of dealing with the topic of racism in discussions.**

|  | 'Scientistic' | 'Pragmatic' | 'Subjectivist' | 'Interculturalist' | 'Critical' |
|---|---|---|---|---|---|
| **Orientation of action** | Objective medical science | Tacit rules of medical practice, experience | Authoritative medical knowledge | Intercultural exchange, cultural knowledge | Sociological scholarship, Self-reflection |
| **Understanding of racialised categorisations** | Given natural categories | Practical classifications | Unclarity, uncertainty | Cultural differences | Social constructs |
| **Understanding of racism** | Intentional, historical | Interpersonal, situational | Relative, subjective | Individual, intercultural | Unintentional, structural |

### Dealing with the topic of racism in a *'scientistic'* way

The 'scientistic' way of dealing with the topic of racism in discussions stands for orientating action towards the idea of medicine as an objective science, justifying the use of racial categories as scientific and defining racism based on intention. This corresponds to the habitus of a medical professional who perceives medicine as an objective scientific endeavour.

The underlying *orientation of action* refers to an account of science and scientific methodology as a resource for explaining and understanding racism. Science is integrated in a professional self-perception of medicine as objective and neutral. These qualities of scientific medicine are believed to enable the use of racial categorisation without being racist: As observable differences between certain groups of humans exist, there is a scientific justification to classify them and consider them in medical reasoning. At the same time, students who deal with the topic in a 'scientistic' way distance themselves from lay perspectives on human differences which are considered more likely to lead to racism, for example due to "wrong intention" (Hannah, 6:38).

This orientation towards a presumed scientific objectivity of medicine is connected to *understandings of racialised categorisation* that include aspects of realism, naturalism, and essentialism: Human races exist and represent natural differences that determine individuals' characteristics. Some medical students argue that humans can be categorised in racial groups by looking at biological features such as physiology or genetics. While one student states that "there is only one race–the human race [...] there are no biological human races, there simply aren't" (Alice, 6:49), another student challenges this idea later in the discussion:

> Earlier it was also mentioned, um that we are all one race of humans and that somehow we don't differ genetically, which I think is only partly correct, because of course we also differ genetically. This aspect of skin colour is talked to death, but um it is still always the first thing that becomes visible and in which groups can be divided. (Hannah, 6:51)

Hannah emphasises that there are natural differences between humans which can be used for categorisation. She refers to genetics and skin colour as criteria for the formation of racial groups. She further elaborates her perspective on races by referring to racial differences in physiognomy that she learned about in her physiotherapy training:

> Um it was about, um well, training as a physiotherapist, and it was about walking speeds. And it was about that you can't equate African-uh-descended people with, for example, Europeans, because they have a different shift of their centre of gravity due to, oh God, different heights of the belly button. So it was about axis ratios [...]. (Hannah, 6:38)

Hannah describes biological race notions as a part of her professional knowledge. She learned about them in the context of her training as a physiotherapist and uses specific

professional terminology to explain that African and European people's physiologies generally differ because of different shifting of "their centre of gravity" (Hannah, 6:38). This scientific terminology lets her views appear as facts, and we gain the impression that Hannah accepts them as scientific medical knowledge. Other students react with criticism. For example, Alice responds to the statement regarding walking speeds by stating that she finds it "somehow problematic [. . .] to make generalised statements about Africa, people of African descent or so" (Alice, 6:41).

Hannah's evaluation of the use of racial categories in medicine points to how students showing a 'scientistic' way of dealing with the topic tend to *understand racism*. She concludes the abovementioned statements by stating that categorising people based on skin colour "wouldn't be racist at all" (Hannah, 6:51). She explains:

> Per se, there is first of all no racist thought behind it, but it's about, um, representing medical conditions somehow, uh, and being able to measure certain things. But of course, um, it can be perceived as racist if it is expressed incorrectly or with the wrong intention. (Hannah, 6:38)

The claim that there is no "racist thought behind it" reflects that intentionality is seen as a criterion for defining racism. It is only "with the wrong intention" that such categorisations might be "perceived" as racist. The word "perceived" reinforces the impression that this approach differentiates between expert and lay perspectives on the subject: lay people might understand such categorisations as racist, but only because they do not understand their scientific legitimacy as representations of objective natural facts. It is therefore not racist to use them for scientific aims, for example to "measure certain things". According to this argument, it is only because of bad intentions or in case of wrong (e.g., non-scientific) use that one should evaluate racial categorisations as racist. A 'scientistic' way of dealing with the topic of racism therefore works as a justification for medical students to talk straightforwardly about assumed medical or biological differences of racialised groups. It appears as an aim to prove the scientific dependability of medicine's racial categorisations.

In the same discussion, Hannah refers to an example of wrong use by addressing the discredited status of racial categories in Germany: "So why is the term race [German: "Rasse"] already racist for many?" And she gives an answer herself: "Because there were once people in history who connoted this term so negatively" (Hannah, 6:51). Hannah explains the racist connotation of racial categories by referring to a misuse in the past. Following her argument, racial categorisations form a neutral taxonomy but due to historical circumstances were "connoted [. . .] so negatively". Julian validates this view later in the discussion and explicitly refers to the German context. He elaborates:

> Um, I could imagine that in Germany this always has a lot to do with the fact that, um, yes, the debate about race, um, yes, is historically very heavily loaded [Nicola nods]. And, um, yes, we still carry a guilt with us, so to speak, at least in the minds of many [. . .], um and yes, we move far away from it, in the German culture, um, from dividing people by race and um, do not like to discuss this topic so openly. (Julian, 6:84)

Interestingly, Julian and Hannah do not explicitly name the historical context that makes it difficult to speak about races or racial groups in Germany. Nevertheless, by referring to the common discourse about German "guilt", Julian implicitly refers to the scientific racism and racial state ideology of German National Socialism that was used to justify racist policies, eugenics, and among other crimes the Holocaust. The fact that the students mostly only

address this topic implicitly–e.g., by referring to "obvious historical reasons" (Alice, 6:85)–
shows that the historical background of racial categorisation in Germany is treated as collec-
tively shared knowledge that one can relate to in the discussions without naming it explicitly.
Julian's speaking about this part of history may convey an impression of discomfort or uncer-
tainty as he uses a lot of fillers and speaks disfluently. Nevertheless, in these argumentations,
racism appears as a problem of the past: Modern medicine's scientific use of racial categories is
perceived as clearly different from these historical faults.

## Dealing with the topic of racism in a '*pragmatic*' way

The 'pragmatic' way of dealing with the topic of racism in discussions stands for orientating
action towards tacit rules of clinical practice, justifying the use of racialised categories as practi-
cal and defining racism as an interpersonal problem. It corresponds to the habitus of a (future)
physician who is seemingly mainly interested in hands-on solutions for the everyday chal-
lenges of clinical care.

The underlying *orientation of action* refers to implicit rules in clinical practice and ostensi-
ble practicability. This way of dealing with the topic of racism also perpetuates racialising
views, but instead of conceptions of science and scientific knowledge, practical considerations
serve as a justification for the use of racialised categorisations in clinical practice. Race talk is
used to justify diagnosis based on racialisation. This becomes apparent in two regards. First,
racism is discussed in the context of the restrictions and problems of everyday health care
practice, for example, an immense workload with high numbers of daily consultations and
lack of time. Second, reflections on racism are orientated towards the perceived practical use-
fulness and reliability of racialised categorisations. In a nutshell, these two orientational aspects
come to the fore in one short sentence: "[U]ltimately, there is actually no other way to do it"
(Julian, 6:36).

A strong orientation towards ostensible clinical usefulness becomes apparent in passages
where *understandings of racialised categorisations* are discussed with regard to the colloquialism
"Morbus mediterraneus", a term that is sometimes used interchangeably with "Morbus bospo-
rus" (Nadia, 4:8; Carla, 4:9) in the discussions. One student explains that the colloquialism is
"about defining an excessive complaining or a theatrical appearance" (Moritz, 1:26), and helps
"to assess the individual expression of pain [. . .] or suffering" (Moritz, 1:28). The underlying
assumption is that there is a connection between a person's "cultural background" and "how
much I, um, show my suffering to the outside world" (Moritz, 1:28; c.f. also: Fenja, 2:19). The
term 'Morbus mediterraneus' vaguely merges notions about culture and geographical origin:
Moritz primarily connects it to "cultural background" while, taken literally, the terms 'Morbus
mediterraneus' and 'Morbus bosporus' refer to specific geographical origins. They are also
understood to refer to "people, who are not European or from Southern European or from the
direction of Turkey, um Syria, etcetera, would be more sensitive to pain and would dramatize
everything" (Nadia, 4:8). In the discussions, it becomes clear that these terms are usually not
explicitly taught in the curricula. Students are confronted with them in clinical rotations when
health care professionals use them, e.g., in "handover" situations (Moritz, 1:26).

Against criticism from other participants, the use of the term 'Morbus mediterraneus' is
defended as handy in the hustle and bustle of clinical practice. Moritz argues that he "consid-
ered it very practical so far" (Moritz, 1:33), "because you don't want to give fentanyl to every-
one who cries out for fentanyl" (Moritz, 1:28). He finds the category helpful in decisions about
pain assessment and treatment, e.g., with strong pain killers such as "fentanyl". Especially in
time-sensitive settings such as the emergency ward, using these racialised categorisations is
considered an efficient means for triage and prioritisation:

> Well, I think that this commitment to the individual always has a very decisive limit, and in medicine that is often very crucial. And that is time. Um, so that's also the case in everyday life, as well as, yeah even more so in medicine, um categorisation and, for example, for something like a triage in the emergency room [. . .] it is very important to be able to classify the flood of patients. (Julian, 6:47)

Julian argues that as a medical professional, one cannot commit fully to the individual patient due to external limitations: Physicians must take care of a lot of patients, so their time is limited. He pictures the situation of a "flood of patients" in the emergency ward that only leaves little time for each consultation and decision. The high demands of everyday clinical practice are believed to call for simple but well-tried shortcuts in decision-making processes, such as 'Morbus mediterraneus'. This is connected to a utilitarian logic according to which the end justifies the means, as "there is [. . .] no other way to do it" (Julian, 6:36). The practical conditions of medical care are used as an argument and defence for employing racialised categorisations in the care of patients, even though it is acknowledged that this practice might "hurt them" (Julian, 6:47). In the tension between considerateness and critical self-awareness on the one hand and efficiency on the other, this way of dealing with the topic of racism habitually favours presumed efficiency achieved by rough racialised categorisations.

Moreover, it is not scientific evidence that is referred to as the primary orientation for clinical practice but "previous experience", "knowledge of human nature" and even "the vernacular" (Julian, 6:36): Medical practice is perceived to draw orientations mainly from experience and common sense. The use of racialised categorisations is evaluated as indispensable because of its previous and common application. As another student emphasises, "Morbus mediterraneus" is "a common term" which "is definitely used also during the handover" (Moritz, 1:26).

Dealing with the topic of racism in a 'pragmatic' way is accompanied by an *understanding of racism* as one interpersonal problem among others: "I think in normal contact between two people, it's almost impossible, yes, to walk together for a long time without stepping on each other's toes once in a while" (Julian, 6:47). This argument is also used to defend the use of racialising categorisations such as "Morbus mediteraneus". It might "hurt" (Julian, 6:47) people to be categorised in this way, but this is framed as inevitable in human interaction and especially in medical practice.

Moreover, the belief that people from different cultures have profound and relevant differences keeps this approach from identifying the use of such racialised colloquialisms as racism. For example, Julian continues his abovementioned statement about the clinical validity of the racialised colloquialism as follows:

> And um, if you relate the whole thing to this Morbus mediterraneus, um, and yes, cultures naturally have different expressions of pain–to what extent is that racism, if you assume that a patient from Southern countries um, expresses his pain faster and more strongly? In contrast to [. . .] the Swede, who, um, yes, in the vernacular probably, um, has a higher pain tolerance. (Julian, 6:36)

Against the backdrop of the idea that "cultures naturally have different expressions of pain", naming such differences is not considered as problematic or racist. Julian assumes connections between geographical origin, culture and how people express their pain. The use of the personification of "the Swede" leads to the impression that all Swedes have "a higher pain tolerance". The 'pragmatic' way of dealing with the topic of racism comes along with generalising assumptions about behavioural characteristics of people from a certain origin or

nationality and uses them as heuristics in the context of clinical decision making. This is perceived as a necessary part of professional health care provision.

This way of dealing with topic of racism is accompanied by a tendency to talk without hesitation, disregarding the discriminatory connotations and possibly hurtful effects of what is said. These effects are even knowingly condoned. Julian, for example, is aware that statements about the "Morbus mediterraneus" and generalising views on people from "Southern countries" or the "Swede" could "hurt" people, but he relativises this aspect by explaining that putting people into groups is "quite natural and time-saving, effective and perhaps also necessary" (Julian, 6:47). Moreover, the use of the specialist tone and technical terminology of everyday clinical practice is characteristic for this approach. This concerns, for example, the use of the medical term "fentanyl" for a specific pain medication (Moritz, 1:28). The reference to real-life health care settings and personal experience conveys an authority connected to the habitus of a physician who practically saves people's lives and health by having a certain routine, knowing more than others, and making difficult decisions and unpopular differentiations all the time.

## Dealing with the topic of racism in a '*subjectivist*' way

The 'subjectivist' way of dealing with the topic of racism in discussions stands for a lack of clear orientation of action and emphasizing uncertainty and subjective assessment when it comes to racialised categorisations as well as racism. Characteristic is the claim of not having enough knowledge about racism to form a clear and definitive position and therefore relying on subjective impressions and opinions.

This way of dealing with the topic of racism is characterised by a general *orientation of action* towards factual definitions, cut-off values and definite criteria as they are commonly learned during medical studies. They provide the habitual framework for evaluations and decision making in the medical context, but their orientational value for dealing with the issue of racism is experienced as limited, which leads to a lack of effective orientation in this context.

Students displaying this way of dealing with the topic describe difficulties with drawing the line where "racism begins" (Hannah, 6:38). The underlying wish for clear definitions and cut-off values remains unfulfilled in discussions about racism as "everyone [. . .] sees the boundaries differently" (Nadia, 4:41). This underlying orientation is illustrated by the wish for an authority that provides a definite definition of racism, a checklist that helps to identify racism, or a guideline that prescribes how to deal with racism. For instance, one student asks "[. . .] if there is somehow a committee that sort of prescribes guidelines for, also for, hmm, so for social aspects of health care" (Franziska, 3:99). By underscoring one's own limited knowledge and perspective, this approach relativises personal responsibility for handling the topic of racism. In order to be able to do something about racism, one first has to learn more about it and gain more certain knowledge.

Regarding the *understanding* of *racialised categorisations*, students showing subjectivist orientations emphasise that they find the topic and its aspects undefined and unclear. For instance, in discussions about racialised groups, there is an uncertainty about which terms, wordings, or categories should be used for those who are affected by racism. Recurring expressions are "not typically German" (Fenja, 2:42), "people with darker skin" (Cathrine, 3:59), "people with a migratory background" (George, 3:101), "marginalised" (George, 3:111), or "foreigners" (Amin, 2:69). Other students resort to terms such as "PoC" (Person of Colour) (Franziska, 3:61) or "BIPoC" (Black, Indigenous, Person of Colour) (Monika, 5:22) to name those who experience racism personally.

Also, regarding the German word "Rasse" [race], one student declares the uncertainties of its use right away: "I'm going to say Rassen [races], even though I know that's not the right term" (Liam, 3:57). He talks about health statistics from the US-context with race/ethnicity categories. There is an uncertainty connected to different meanings and connotations and to the translation of the word "race" with the German word 'Rasse'. By being transparent about knowing that it might not be the adequate wording, Liam is taking a performative distance from the moral implications of using the word 'Rasse' and indicates that he *actually* does *not mean* it. Thereby, possible critical interpretations of his statement are put into perspective in advance.

Furthermore, students who show a 'subjectivist' way of dealing with the topic articulate ambivalence about their *understanding of racism*, as well as about its relevance and moral evaluation. From a subjectivist standpoint, students argue that their own perspective is ambivalent and uncertain because of a lack of knowledge. While a 'pragmatic' way of dealing with the topic tends to rely on a narrower definition of racism that prevents the moral condemnation of phenomena that other approaches would clearly understand as racist, a subjectivist perspective exhibits an unclear or ambivalent moral positioning. For example, Ben says about the term "Morbus mediterraneus": "I don't know to some what extent [sic] that's racist, but at least that's how I perceived it" (Ben, 1:24). Here, an uncertainty pervades Ben's evaluation of the term as racist. The malapropism "some what extent" [German: "irgendwieweit"] fuses the words 'somehow' and 'to what extent'. This implicitly conveys an ambiguity and insecurity about whether his perception of the term as racist would be shared by others. He highlights a perceived uncertainty in understanding and morally evaluating racism by pointing to his subjective impression: "at least that's how I perceived it". According to this way of dealing with the topic, defining racism mainly depends on subjective perspectives and experiences.

Taken together, these aspects may work to mitigate one's own responsibility. Students describe it as their motive for participating in the group discussions to learn more about racism and to gain a clearer moral orientation. In his first statement, Moritz shares: "I think I'm here because I'd like to orient my own compass and see how much discrimination, racism, etc. I experience myself and how much of it can I tolerate and how much not?" (Moritz, 1:4) In this statement, a focus on personal opinion in dealing with racism becomes apparent. For Moritz, the discussions are useful to "orient" his "own compass" and to see how much racism can be "tolerated" from his subjective perspective. Stressing a not yet orientated moral compass about racism can be interpreted as a way of relativising one's personal responsibility and the possibility to be held accountable for how one discusses and approaches the topic of racism. Statements and positions that are criticised by others might always be excused to be part of one's learning process or due to a lack of objective definitions of the phenomenon of racism.

A 'subjectivist' way of dealing with the topic of racism is connected to a moral self-referentiality of talking about racism. Some medical students make an effort to dispel the impression that they might behave in a racist way. For instance, Ben vehemently points out that he has never used the term 'Morbus mediterraneus' himself but observed others using it. When another participant explicitly calls the term "super racist" (Susan, 1:25), he reacts: "I mean, I have never had the opportunity to have to use this term in any way or not to want to use it, but I had noticed that" (Ben, 1:32). For some students, talking about racism is connected to the unsettling question if they might have reproduced racism themselves in the past or might act or say something "wrong" (Rose, 5:55) in the discussions. By emphasising uncertainty and the need for more reliable information, a 'subjectivist' way of dealing with the topic of racism can function as a strategy to reduce the moral burden of this thought.

## Dealing with the topic of racism in an '*interculturalist*' way

The 'interculturalist' way of dealing with the topic of racism in discussions stand for orientating action towards an ideal of intercultural exchange, understanding racialised categorisations as representing cultural differences and racism as prejudice against cultures. The corresponding habitus is that of a worldly and cosmopolitan medical student who thinks that people must talk more with each other to understand cultural differences and leave prejudices behind.

The underlying *orientation of action* refers to intercultural exchange as the solution for racism in medicine and health care. It is connected to the implicit assumption that racism is a characteristic of uneducated people or societies. The idea that the acquisition of knowledge about "other cultures" (Fenja, 1:55) minimises racism gives the impression of a linear progression away from uninformed racist manners towards informed and interculturally enlightened medical professionals. Therefore, integrating aspects of "cultural understanding" (Fenja, 1:55) in medical education is perceived to be an important starting point for tackling racism.

Interculturalist *understandings of racialised groups* are not about biological or physiological features, but rather about cultural differences. In a discussion about the definition of racism, one student explains:

[. . .] I wouldn't associate it with the term 'race' ['Rasse'], for me. But rather that one somehow constructs groups that then have a shared cultural origin or religion or perhaps also skin colour, um, so they have a commonality there. (Lara, 6:42)

Lara stresses that for her, racism has not so much to do with the German term 'Rasse' but with group constructions that are primarily based on culture or religion. The construction of such groups is understood to be based on their "commonality".

Accordingly, students who deal with the topic of racism in an 'interculturalist' way explain ethnic differences in mortalities within the context of the COVID-19 pandemic in the USA through cultural specificities: "and, um, yes, with Hispanics, of course, the family is very important and, um, yes, simply also dealing with the disease itself" (Julian, 6:72). Julian argues that Hispanics showed the highest rate of excess death during the COVID-19 pandemic because they prioritise family or live in bigger families. For him, being Hispanic means a specific way of "dealing with the disease itself". We observed similar patterns of essentialising arguments about culture influencing how people *are* when it comes to pain, as students find that "in many, um, cultures pain is expressed differently" (Fenja, 2:19).

This is similar to the use of racialised categorisations supported by the 'pragmatic' way but with different underlying orientations. This becomes apparent when the impression that providing health care "often becomes complicated" due to "non-comprehension" (Fenja, 2:19) is addressed in an 'interculturalist' way. While a 'pragmatic' way of dealing with the topic of racism tends to accept cultural stereotypes as rough but handy tools for medical practice, students pursuing an 'interculturalist' way want medical staff to go beyond prejudices and to foster "cultural sensitivity" (Mira, 4:97; Marie, 4:96) with culturally adapted health care for overcoming "non-comprehension" and inadequate care.

*Racism is understood* as a problem that arises from a lack of exchange between members of different racialised or culturalised groups. Because of this, one student explains that she finds it important to address situational racism in the presence of the person who is discriminated against, as this would help to overcome racism. She argues:

so [. . .] this racism comes from somewhere and perhaps also from the fact that not enough dialogue takes place between, let's say, the one race and the other [. . .] [in German: "zwischen [. . .] der einen Rasse und der anderen"]. (Lisa, 5:43)

Later Lisa explains that dialogue "between the one race and the other" is needed to "clear up prejudices" (Lisa, 5:51). Her understanding of racism emphasises prejudices as a central aspect. The idea of intercultural exchange as a means for addressing racism finds support from other participants in the group discussions. Despite the vehement criticism that Lisa faced after proposing more dialogue between "one race and the other" ("I actually just find that very, very, very problematic [. . .]" (Pia, 5:46)), another student supports the idea of bringing people from different cultural groups together for addressing interpersonal racism in teams of health care professionals:

But for me, this point of cultural understanding was a little bit the starting point why I said, okay, maybe you should bring the people together and let them talk to each other. Um, it can also be the wrong approach. But that's a bit of the reason why you send students on an exchange, to get to know other cultures and to understand them [. . .]. (Rose, 5:55)

The student confirms the importance of intercultural exchange and understanding in the context of racism. People with prejudices should "talk to" people from the "other cultures" to lose their prejudices. She relativises her enthusiasm about this approach after other students express different views but nevertheless argues in favour of an 'interculturalist' understanding by referring to school exchange programs.

In other group discussions, students also prominently voice the need for more "cultural competence" (George, 3:111) or "sensitivity to different cultures" (Marie, 4:96). They repeatedly demand that such competencies and knowledge be taught in medical schools:

Because of the lack of knowledge about other people or people of other origins, that you could also learn socio-cultural basics in, in medical school, that [. . .] one is sensitised again that there are different cultures and that all cultures, so to speak, have their individual, um, expression of illness and pain [. . .]. (Maya, 2:156)

Maya identifies a "lack of knowledge" in the medical professions about "other people or people of other origins" and demands medical schools to teach specific knowledge about other cultures such as specific "expressions of illness and pain". The discussion about racism hereby focuses on cultural differences and knowledge about these differences appears as the medical professions' antidote to racism. As other students criticise the use of the term 'Rassen' in her plea for more dialogue between racialised groups–"there are no various different races, [laughs shortly] between which one can mediate. There just aren't" (Paula, 5:50)–Lisa later reacts with the wish to change the word 'Rasse' to 'ethnicity' in retrospect:

And at that moment, didn't think about the fact that this is of course a word, um, that in any case one should never use in such a situation. [. . .] I wanted to replace the word with the word 'ethnicity' in retrospect. I hope that you can all accept that in no way it was meant in a negative way in this context. (Lisa, 5:51)

Lisa acknowledges that using the word 'race' can be problematic but restricts her re-evaluation to "such a situation" which might imply that there are other situations that allow for an unproblematic use. Instead of explicitly allowing for insecurity about which term might be

right, she presents the idea of "replacing the word ['race'] with the word 'ethnicity'" (Lisa, 5:51) as a solution. This idea of exchanging one word for the other illustrates the underlying view on racism as a problem of intercultural relations, and on racialised groups as representing different cultures.

The 'interculturalist' way of dealing with the topic of racism in discussions is often accompanied by a specific self-perception as being above racist or prejudiced thinking. By explaining to other participants how situations of racist discrimination should be addressed, and that racism is a matter of knowledge deficits, this approach conveys the impression that racist thinking can be overcome once and for all. Corresponding suggestions seem to rely on a point of view that has left behind racist, prejudice-laden thinking: "Um, if you say to these people now, okay, why don't you just go directly to the person concerned with your problem and explain to them what they did wrong?" (Lisa, 5:43). "These people" who are acting in a way that Lisa perceives as racist should be told to "go directly to the person" to overcome their underlying prejudices. Indeed, the other ways of dealing with the topic are also associated with the notion of being above racism, but instead of deriving this self-perception from scientific expertise or good intentions, the 'interculturalist' approach emphasises the need to acquire knowledge about different cultures, e.g., via intercultural exchange.

## Dealing with the topic of racism in a '*critical*' way

The 'critical' way of dealing with the topic of racism in discussions stands for orientating action towards sociological scholarship, understanding racialised categorisations as social constructs and viewing racism as a structural problem. This corresponds to the habitus of a critical and politically interested student who takes distance from what is generally perceived as normal medicine and health care in order to identify and criticise inherent racism from a broader perspective on society and the health care system.

The underlying *orientation of action* refers to ideals of anti-racism, critical awareness and (self-)reflection. Critical knowledge about racism as a structural phenomenon and a confident moral orientation serve as distinguishing characteristics from other students and medical professionals. Reflection on positionality regarding racist power structures is perceived as an important aspect of dealing with the topic of racism. There are students who reflect on different social positionings, and how they influence one's chances in life and the burden of discriminatory experiences: "But I'm also aware that it's actually an expression of my privilege [Rose nods] not to have to deal with it all the time, and that I want to find ways to deal with it in solidarity" (Monika, 5:49). Monika considers not experiencing daily racist discrimination a privilege that she wants to use for standing in solidarity with people who are directly affected by racism.

The corresponding *understanding of racialised groups* refers to insights from social sciences or cultural studies. Racial categorisations are perceived as historical products of racism that have their roots in the justification of colonialism:

> I mean, in principle, there are no human races biologically, that simply does not exist. And, um, races were actually invented by white people, in order to actually justify colonial imperialism and to, yes, excuse it. (Alice, 6:41)

In consequence, students problematise the use of racial categorisations. They refer to sociological concepts such as "othering" (Susan, 1:15) or "racialisation" (Alice, 6:52) to explain the formation of racial groups:

> I think racism is about racialisation. And, yes, as I said, it is not the same as race, but they are just fuzzy characteristics. So it's also, it's also mainly external characteristics. But there are also cultural characteristics. (Alice, 6:49)

Alice identifies "racialisation" as a fundamental aspect of racism that refers to the construction of racial groups based on external as well as cultural attributes. Moreover, this way of dealing with the topic often aims for the argumentative deconstruction of racial groups–both regarding biological (Alice, 6:41; Alice, 6:52) as well as cultural notions (George, 3:71). For example, Alice argues against the notion of biological races by stating that "there is more genetic diversity among Black people than between Black people and white people" (Alice, 6:41). She calls racialisation "arbitrary" (Alice, 6:49).

We see that a 'critical' way of dealing with the topic of racism can be associated with the use of words that are uncommon to the medical domain. Terms and concepts from social and cultural sciences are employed to explain the functioning of racism and to criticise the use of racial categories in health care. This kind of theoretical knowledge is partly taken for granted, for instance when races are called a "construct" (Alice, 6:49) or the result of "racialisation" (Lena, 2:140). In a similar vein, the concept of othering is implicitly referred to when Susan talks about the term 'Morbus mediterraneus' and says: "that's exactly the categorisation so for other people" (Susan, 1:25).

On the other side, medical and biological arguments against racism are brought to the fore, for example, by questioning the idea that "discrimination is efficient" (Susan, 1:46) or by pointing to the biologically unprecise character of racial categorisations (Alice, 6:41). Accordingly, the necessity to be aware of the influence of racialisation and structural racist discrimination on health is highlighted–"if left out, this would actually be colour-blindness" (George, 3:101). Therefore, the critical use of racial or racialised categorisations in medical research is considered important for understanding and then fighting the effects of racism on health:

> But of course, there are very big differences in treatment and things and access to education and, um, access to housing, access to jobs. [. . .] And we have to make that visible as long as it exists so that we can do something about it. (Alice, 1:120)

Students displaying a 'critical' way of dealing with the topic oppose personalist and situationist views of racism that are articulated in a 'scientistic', 'pragmatic', or 'interculturalist' way. They express an *understanding of racism* that stresses unconscious and structural aspects and considers the consequences for those who are negatively racialised. Some even understand racism as anything that results in a disadvantage for racialised groups, which means that "the impact is more important than the intention" (Alice, 6:41). Structural understandings are connected to reflections about power and social positions, as racism "has to do with power position, um, that is a power relationship between people who have resources and people who have rather less resources" (Maya, 2:96). Moreover, racism is identified in the health inequalities that exist between racialised groups and in the negative effects of structural racist discrimination on health: "You can see that racism is a problem because, for example, Black people or People of Colour have higher mortality rates, which is a very drastic consequence of racism, so to speak" (Lena. 2:119). Another student emphasises: "Racism kills people in medicine" (Pia, 5:102). There is a clear moral disapproval connected to this understanding of racism. Accordingly, some students explicitly denounce the moral misconduct of others reproducing or tolerating racism.

Social inequalities between racialised groups are understood as the effects of structural racism which do not necessarily have to be intended. Racism is "something that works

unconsciously" (Nicola, 6:45). Students express the idea that in the discourse about racism, "this personal question of guilt should be excluded to some extent" (Monika, 5:57). Some students consider personal intent mostly irrelevant for identifying and evaluating racism: "[I]n the end, the vast majority of people, when they behave in a racist way, have no bad intention at all", one student states and clarifies: "[R]acism is so widespread in our society that I have also behaved in a racist way and probably still will" (Pia, 5:53). Situational racism, prejudices and racist behaviour in interpersonal interactions are interpreted as symptoms of collective societal or structural racism. Yet, students who deal with the topic in a 'critical' way are sensitive to and critical about racialised prejudices and racist discrimination in health care. Generalising views on racialised patients and their negative effects on health care are condemned (Mira, 4:36).

Following this moral take on racism, students dealing with the topic in a 'critical' way tend to explain or teach others about racism and its functions. However, this mode of discussing racism can prevent mutual discussions from taking place. For example, Alice explains several things about her perspective on racism and deconstructs essentialist notions of racialised groups. Later in the discussion, she asks, "if people can't understand [me], um, or doesn't anyone go into the things I say because they don't make sense?" (Alice, 6:55) She expresses her uncertainty about whether her communication worked, and whether she was able to get her points across clearly because other participants in the discussion did not react on what she said but primarily referred to other students' statements. She considers her way of argumentation as a possible reason and explains: "I just wanted to get feedback on my tone" (Alice, 6:58).

Sometimes a 'critical' way of dealing with the topic of racism is even connected to a tendency of showing off critical reflections while distancing oneself from other people who are perceived to be not as reflected about racism. One student dissociates herself vehemently from people who use racialised categorisations such as 'Morbus mediterraneus' in clinical practice. She finds it "unbelievable, that this exists" (Susan, 1:25) and "completely crazy" that other students "have heard" it–she has only ever <u>read</u> about it". She shares how she reacted when she read about the term:

> Yes, is super racist [laughs] I'd say. [. . .] But I couldn't believe it at all. I thought they were completely insane [laughs briefly], that they use something like that [laughs again]. Um, yes, so one really first has to, one first has to come up with the idea of categorising people like that and then inventing an extra word for it, yes. (Susan, 1:25)

Unlike the 'subjectivist' way, which highlights the ambiguity in the evaluation of racism, Susan's evaluation is presented as unimpeachable. Her insistent clarification about the wrongness of employing 'Morbus mediterraneus' in medical practice directly responds to Ben's statement who expressed that he is not sure to "some what extent [sic] that's racist" (Ben, 1:24). Susan firmly locates the term's use in the sphere of other people's behaviour and dissociates it from her own acting and thinking. By calling it "super racist" (Susan, 1:25), she clearly identifies 'Morbus mediterraneus' as a racist term for herself. Her laughing reinforces the impression that Susan distances herself from others' (more uncertain) perspectives. It also conveys the impression that she embodies an external perspective on medical practice. For her, using such a categorisation in practice is unbelievable, and she criticises her colleagues for doing it and not being sufficiently reflected: Physicians "see themselves so super great as demigods and are very exoticising and racialising and [. . .] very few people notice it and talk to them about it" (Susan, 1:40). By emphasising the fact that she has only ever <u>read</u> about it, she distinguishes herself from practitioners who use the term without consideration. She appears as the expert who reflects on the practice critically from a more rational and educated point of view, and

who is able to explain to other medical students or staff why their behaviour is problematic with regard to racism.

## Discussion

Our analysis of the group discussions reveals the scope of medical students' ways of dealing with the topic of racism in discussions. The five typical approaches we identified are each connected to certain habitual orientations of action acquired in the medical context through professional socialization. They all involve different understandings of racialised categorisations, racism, and the relation between both. Each of them also shows specific difficulties in the discussion of racism.

By referring to aspects of the medical profession that are generally evaluated as positive, the 'scientistic' and the 'pragmatic' way of dealing with the topic tend to normalise racism in medicine and health care. They emphasise the scientific background or good intentions of medical professionals that aim to provide the best patient care, but thereby legitimise racist practices [17]. Also, these ways of dealing with the topic obfuscate how essentialised views on racialised groups may reproduce racism [71]. Especially the 'scientistic' way also provides empirical examples for the "relocation of racism to the past" [123].

Regarding the relation between racialised categorizations like 'race' and racism, the 'scientistic' perspective perceives 'race' as a category referring to naturally given groups and interprets racism as an ignorant or malevolent misuse of this category. This perception of racial groups is quite common in Germany: Half of the population (49%) believes in the existence of biologically distinct human races ("Rassen"), but this belief is more common among people with lower levels of education [43]. It is therefore important to investigate why the idea of biological races can appear persuasive in medicine and health care while the term 'race' is avoided and sometimes conflated with other categories such as culture, ethnicity, and migration [43, 124, 125]. Instead of pursuing a "race evasiveness", Aikins and colleagues (2024) propose to strengthen a biosocial approach that widens the perspective of medicine on these categorizations by acknowledging the interdependencies of genetics and racism [124].

Similar to the 'scientistic' way of dealing with the topic, the 'pragmatic' approach does not perceive the racialised categorisations that are used in clinical practice as a result of racialisation, racist prejudices and bias, but as handy heuristics facilitating clinical decision-making. The reference to cultural differences instead of biological 'races' illustrates the semantic flexibility of racialisation that not only constructs biological differences but, in the sense of a "racism without races" [126], employs the notion of homogeneous 'cultures' or 'ethnicities' in order to create groups that can become the reference point of racist discourses and practices. Kattmann (2017) therefore calls racism and culturalism "sisters in spirit" [125].

The 'subjectivist' way of dealing with the topic of racism in discussions leads to problems because the objective existence of racism is called into question when habitual expectations for clear-cut textbook knowledge, criteria and definitions are not met. Additionally, the empirical findings connected to the 'subjectivist' approach point to several difficulties that medical students encounter regarding the understanding and use of racialised categorisations such as 'Rasse'/'race'. Problems connected to the wide range, vagueness and inadequacy of human categorisations are not specific to medical students but are also described for the general context of life sciences in Germany [127]. Regarding the relation of racialised categorizations and racism, the use of racialised categorisations such as 'race' or 'Morbus mediterraneus' is seen as somehow problematic and vaguely associated with racism. However, due to uncertainty about what racism is, who is affected by racism and how to deal with racialised categorisations in the clinical environment, the evaluation of the whole topic is afflicted with uncertainty and

subjectivism. This mirrors knowledge gaps among medical students about the concept of racism and other forms of discrimination [97, 102, 121].

The 'interculturalist' way of dealing with racism is connected to a simplified view on the topic in terms of intercultural misunderstandings. Students who deal with the topic in an 'interculturalist' way suggest that acquiring more knowledge about the special habits and needs of those who are not perceived as members of the 'normal' patient group provides a solution to racism in health care. This is based on essentialised distinctions between cultures and can lead to the implementation of racialised knowledge into professional practice. Nevertheless, it may find broad support as more knowledge is generally perceived as a good characteristic of health care professionals. But, as Philomena Essed points out with regard to concepts of "multiculturalism" [55] and "cultural tolerance" [128], such approaches "lack the ideological and practical tools to address the structural inequalities" [55]. Moreover, the 'interculturalist' way of dealing with the topic shows tendencies of a "culturalization" that makes racism "invisible" because racist differentiation can happen in "a decent way" without reference to race or racial categorisations [123]. This may be reinforced by the 'interculturalist' understanding of the relation of racialized categories and racism. Race categories are perceived as unimportant for racism. Instead, cultural groups are seen as holding essential characteristics that one should learn about to avoid being racist. Thus, racism is not identified as the underlying ideology that leads to essentialising views on cultural groups. By understanding racism only as prejudice about cultures, the idea of cultures may become homogenising and essentialising.

Finally, the 'critical' way of dealing with the topic of racism faces difficulties because of the habitual outsider perspective adopted in order to develop a critical view on structures and practices in medicine and health care. While this approach might actually enact what Kendi calls the crucial part of anti-racism, that is "consistently identifying and describing it [racism]–and then dismantle it" [53], we observed that some anti-racist ideas ended up seemingly isolated in the group discussions as other students appeared to have difficulties following and understanding their ideational foundations, or to react to perceived moralizing condemnation of racist behaviour with denial or defensiveness [108]. Regarding the relation of racialised categories and racism, the 'critical' way of dealing with the topic is the only one that located racism at the root of racialised categorisations such as the concept of race and acknowledges its connection to the historical contexts of academic racism, imperialism and colonialisation.

The implications of these results should be discussed in the wider context of research on racism in medicine and health care. Thus, several studies have illuminated health care professionals' essentialised views on racialised groups. They show that essentialised and biologistic understandings of 'race' are perpetuated in medical schools: Lecturers talk about 'race' as an independent risk factor for diseases without contextualising the term or explaining that 'races' are not biological entities but social constructs [97, 129]. Our research indicates that some medical students are open to including such essentialised views on racialised groups in their clinical reasoning. The literature points to consequences of such racial bias, e.g., on diagnostics, physician-patient interaction and treatment. In Germany, a recent qualitative study found that racialised bias about patients' sexuality may lead to over- and underdiagnosis of sexually transmitted infections (STI): Black women are hypersexualised and more likely to be checked for STIs while Muslim women experience medical professionals to be more hesitant to propose such testing [49]. Here, cultural essentialism seems to intersect with racism and other forms of discrimination such as sexism and may lead to inequalities in health care. Moreover, undertreatment for pain in Black Americans is associated with medical students' and physicians' false believes about biological differences between black and whites [70].

Regarding pertinent research on racial bias, our findings highlight how medical students' ways of dealing with the topic of racism result from complex connections and negotiations of explicit and implicit orientations. For example, 'scientistic' arguments show that medical students hold explicit biased views and feel encouraged to use them as clinical tools, especially in situations of high workload. Thus, not only implicit bias could increase due to stress [87], but also the willingness to apply biased clinical shortcuts in the treatment situation, especially if they are not recognised as racism. Here, our results underscore the influence of professional socialization and habitus on medical students' ways of dealing with the topic of racism. Another focus group study conducted in the UK shows that medical students were more at ease to talk about race in the context of a biomedical model which emphasises genetics or susceptibility to diseases [48]. Our study indicates that professional socialisation in medicine might promote this preference: Medical studies tend to prefer specific modes or topics of discussion about race (e.g., biomedical, pathophysiology, risk factors) over others (e.g., social dimensions of health, inequalities) due to spending more time with the former or learning from biased role models [74, 97, 129]. Our results further differentiate such findings by proposing that relative to the underlying habitual orientation, medical students might be more or less inclined to discuss racial aspects as biologically relevant for health care. In this regard, Bourdieu's concept of habitus promises a more differentiated understanding of the connections between implicit biases as cognitive correlates of acquired dispositions and their roots in social structures [130, 131]. The concept can thus help avoid simplified and individualized views on racism in medicine and health care by highlighting the influence of structures and collective orientations on individuals' ways of dealing with racism.

Furthermore, our results can help to understand difficulties that may arise when different ways of dealing with the topic interact in medical students' discussions of racism. Opposing or even conflicting orientations can lead to controversial discussions. However, they may also impede the development of a collective conclusion or even the achievement of a mutual understanding. Discussions then take more parallel patterns of shared opinions and lack true interaction between participants [120, 122]. In our groups, we observed that some medical students were not used to being considerate regarding potentially discriminating or devaluing effects of their statements on people who personally experience racism. In particular, this holds true for objectivistic views on racialised groups like the ones promoted by 'scientistic' arguments, or the handy heuristics employed in the 'pragmatic' approach. The somewhat naïve do-gooder-attitude of the 'interculturalist' perspective can also effectively prevent critical self-reflection and a considerate habitus in discussions of racism. Such ways of dealing with the topic often caused indignation and were vehemently criticised by other participants in our groups. However, there were also insecure students who were self-conscious and had difficulties expressing their thoughts about racism out of fear of saying something wrong or hurting somebody. Especially in interactions between 'subjectivist' and 'critical' attitudes, the inhibitions of the former can be intensified. Students displaying a 'critical' approach sometimes claimed for themselves a superior moral ground and referred to unconscious and structural aspects of racism that are often beyond the reach of individual (self-)awareness and intentional change. This way of dealing with the topic of racism can intimidate other participants in discussions. Yet, there might also be a productive aspect in the irritations that arise when habitual ways of dealing with racism collide. Especially the 'subjectivist' wish to learn more and reduce uncertainty and ambivalence about the topic of racism points to the potential of anti-racist medical education.

The proposed characterization of medical students' approaches to the topic of racism in discussions adds new dimensions to pertinent research in the context of medical education. For example, while Novak and colleagues (2022) describe the first stance of "passive adaption" as the stance where students show no intention or motivation to address the topic of racism, our

study indicates that there are different habitual aspects that prevent medical students from acting against racism. Regarding the highest "stance" identified by Novak and colleagues, we found that acknowledging a personal responsibility to act against racism is only one important aspect for enacting anti-racism. Our results make clear that habitus and knowledge aspects also play an important role for what is understood as anti-racist action. Thus, the analysis of the 'interculturalist' way illustrates how even well-meaning intentions may lead to behaviour that can effectively perpetuate racist stereotypes or attributions.

## Reflexivity, limitations, and validation

In line with the overall perspective of this study, it is important to reflect on the influence of our own habitus as white researchers on our research practice with regard to social, disciplinary and scholastic aspects structuring our work [132]. Among other things, our perspectives on the topic of racism and on non-white racialised groups are shaped by our white positionality, by our socialization in academic (SG, MS) and clinical (SG) medicine as well as medical ethics (MS, SG), and by the specificities of the German discourse on racism and race. We reflect possible influences of these aspects on our relation to the field, the studied groups, the topic, and the overall findings [132–134].

As a medical student, SG conducted this research in a field and on a group he knows from personal experience: medical students in medical schools. He experienced the absence of the topic of racism during his own medical studies and the difficulties of engaging in discussions about racism with other students or health care professionals both in academic and clinical contexts. As a university professor of medical ethics, MS learned about the lack of anti-racist teaching activities during a brainstorming session held as a participatory format for curriculum development with medical students. He knows the field and medical students from his teaching activities in medical schools. Our access to the field and recruitment of participants benefited from both SG's connection to medical students at various universities in Germany as well as MS's professional network among medical ethicists used for distributing the call for participation.

Being white raised the question of how we as white student/faculty can implement effective anti-racism teaching that also addresses institutional and structural aspects without reproducing racism. Impressions from clinical health care practice brought in by SG indicated the need to study not only medical students' theoretical views on racism but also their ways of dealing with the topic in practice-related situations such as discussions. The aims of our study were shaped by our perspectives as a white medical student and as ethicists in medical education because we are concerned with understanding the preconditions and obstacles of teaching and learning about racism and anti-racism in medicine and health care and their connections to professional socialization in the medical context. We studied perspectives of diverse students with and without personal experience of racism and their interactional dynamics in discussions laden with power relations stratified by racism.

Despite the precautions we took to reduce the risk of exposing the participating non-white racialized students to reproduced racism in the group discussions (see methods section), the researchers' white positionality might have inhibited people who experience racism to articulate their experiences or–conversely–encouraged those students who did not experience racism to share their perspectives [135]. To control for this limitation, group discussions with more homogeneous compositions or one-to-one interviews might be considered. We did not employ such methods for data collection because our research questions did not include a comparison of how medical students with vs. without personal experiences of racism deal with the topic. Instead, this study focused on possible ways of interaction that develop in discussion settings among students with diverse positionalities.

The specific discourse on racism in medicine and health care in Germany influenced our hesitation to collect data about racialisation among our participants. The use of race categories is not common in social and health care sciences in Germany [124, 125, 127, 136]. We included an open question in the prequestionnaire that allowed for self-identification and voluntary disclosure about experience of discrimination. The choice to use the concept of *racialisation* despite all the critique that this concept faces reflects our attempt to take into account the public and scientific discourse on "racism without races" in Germany and Europe [126].

Our results must be interpreted with further limitations of our study in mind: Our small sample size and qualitative approach do not allow for generalisations. Studies with bigger samples and representative surveys of medical students' perspectives on racism in medicine are needed. Moreover, the online setting reduced the possibilities to collect data on body language and other non-verbal communication that might have been useful for a more detailed analysis of habitual aspects. Eventually, our distinction of five typical ways of dealing with the topic of racism might need further differentiation or expansion. For example, our material suggests that there may be more than one 'subjectivist' and 'critical' approach among medical students.

Still, following Yardley (2017) [137], we can identify important factors strengthening the validity of our qualitative results. Thus, the high context-sensitivity of our data regarding medical education in Germany provides localised knowledge about the scope of medical students' ways of dealing with the topic of racism as well as conceptual and linguistic specificities. For example, our data highlights different understandings of racism and race and other racialised categories, such as the biased colloquialism of "Morbus mediterraneus", for the specific German-speaking context of health care. Furthermore, the inductive procedure took the qualitative material as a starting point for an extensive qualitative analysis that structured content with regard to thematic aspects [119] but also developed more depth by adding a detailed analysis of the underlying habitual orientations of action that could be analysed in meta-linguistic aspects of group discussions such as the sequence of statements and paralinguistic reference to shared tacit knowledge, e.g., about the Holocaust [138]. Rigorous comparison of different orientations of actions resulted in the development of an inductively reconstructed typification [30]. To enhance transparency, our manuscript complies with the COREQ criteria [139] for reporting qualitative research and provides the reader with detailed quotes from the empirical material to make our interpretations traceable. Finally, the practical importance of our results lies in the potential usefulness of the provided typification for the further evidence-based development of anti-racist teaching material and anti-racism strategies for students and professionals in medicine and health care.

## Conclusions

Our results allow several conclusions regarding the requirements of implementing anti-racism into medical practice via medical education. The variety of different ways of dealing with the topic of racism in discussions suggest that there can be no didactical one-size-fits-all solution [47]. Medical education needs to tackle racism in different ways and with different foci to foster anti-racist approaches.

For example, ways of dealing with the topic that tend to normalise racism may require a focus on teaching knowledge about racism and competencies that help to identify racism in practice and in internalised habits and thoughts [108]. This might pose particular difficulties as thorough self-reflection is required, and habitual perspectives on racism and the moral evaluation of one's own habits may have to change. The 'subjectivist' way may be addressed by providing access to knowledge about racism, and by empowering critical reflection to overcome feelings of ambivalence and ambiguity connected to the topic of racism. The 'critical' way might benefit from expanded knowledge and discussions about racism.

Being transparent with the existent ambiguities that are connected to discourses about definitions of racism and anti-racism might support an understanding of different approaches to racism while key points of action can be identified, e.g., improving the anti-racist analysis of routines in medicine and health care, developing strategies about practical enactment of anti-racism, support for self-organisation and empowerment. In this context, the literature, e.g., regarding the integration of anti-racism in the CanMEDS competency framework [3], refers to Paolo Freire's concept of "Critical Consciousness" that emphasises peoples' ability to act for change [112]. In addition, medical schools and medical education need to incorporate more interdisciplinarity, especially including perspectives from social sciences and cultural studies, to foster discussions about racism and the analysis and understanding of structural aspects that impede good health care for all.

The interactional difficulties connected to diverse habitual approaches to racism might be addressed by efforts to create discursive spaces that allow for learning while acknowledging potentially different needs in discussions about racism [107]. Metacommunicative agreements might help to establish a constructive atmosphere of discussion. Teachings on how giving feedback on each other's behaviour regarding racism is an important aspect to create a constructive environment that supports self-reflection and on the long run collective changes of habitualised thought and action [140]. Moreover, the habitual dimension of approaches to racism calls for didactical methods that help to ingrain anti-racism in medical students' professional and personal habits, e.g., learning from role models, emphasising critical reflection competencies and value education [141].

Eventually, the difficulties we observed on a habitual level in how medical students deal with the topic of racism point to more general structural problems. These go beyond the scope of didactical approaches in medical education and therefore require more fundamental and encompassing changes in the medicine and health care system itself [75, 142]. In this sense, the enaction of anti-racism must ultimately transcend the teaching context and also address the 'hidden curriculum', the institutional framework and materialised context that are shaped by racism as well as crude scientific positivism, authoritarian traditions and hierarchical power structures connected to a professional culture often lacking emotional awareness and self-criticism.

## Supporting information

**S1 Text. Discussion guide.**
(DOCX)

## Acknowledgments

We would like to thank Merle Weßel for her important work in the design and realisation of the research project and Houda Hallal for her helpful comments on the manuscript. Moreover, SG is grateful for the feedback that he received on his presentation at the conference "Rassismus, Diskriminierung und Gesundheit" in Berlin, March 2023. Many thanks to Lucas Rateitschak for his help in the transcription process and to Emma Dierlamm and Maya Heins for language editing.

## Author Contributions

**Conceptualization:** Simon Matteo Gerhards, Mark Schweda.

**Data curation:** Simon Matteo Gerhards.

**Formal analysis:** Simon Matteo Gerhards.

**Investigation:** Simon Matteo Gerhards.

**Methodology:** Simon Matteo Gerhards, Mark Schweda.

**Project administration:** Simon Matteo Gerhards, Mark Schweda.

**Software:** Simon Matteo Gerhards.

**Supervision:** Mark Schweda.

**Writing – original draft:** Simon Matteo Gerhards.

**Writing – review & editing:** Simon Matteo Gerhards, Mark Schweda.

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
