## [Decision Letter · Decision Letter 0]

13 May 2024

PONE-D-23-43307Medical students’ habitual approaches to discussions about racism in Germany: A qualitative analysisPLOS ONE

Dear Dr. Gerhards,

Thank you for submitting your manuscript to PLOS ONE. After careful consideration, we feel that it has merit but does not fully meet PLOS ONE’s publication criteria as it currently stands. Therefore, we invite you to submit a revised version of the manuscript that addresses the points raised during the review process.

The manuscript has been carefully assessed by myself and an anonymous reviewer. We generally share the same concerns about the manuscript in its current form and invite you to address the following issues in a major revision: - Tightening and improving conceptual definitions. As much as possible, it would be good to be judicious in invoking concepts related to race and racism. The manuscript may currently be trying to address too many ideas at once, making it conceptually unwieldy. Stick to a more selective subset and offer more precise definitions for the ones which truly matter to the argument.- Enhancing correspondence between analytical claims and evidence provided. Various points are raised in the analysis which do not empirically match the data that is given and unpacked. Articulate insights with more explicit connections between data and interpretation to strengthen the results. More detailed discussion of how the present insights link to broader scholarship and inquiry on race outside the medical field -- or perhaps other forms of inequality/inequity in the medical field -- will also be ideal.- Expanded validation and reflexivity statement. Qualitative analysis is shaped by subject positions of analyst/s and these are especially salient in power-laden domains such as race and racism. Improved discussion of the authors' relationship with the topic, participants, and overall findings will be critical for this manuscript. Additional discussion of how themes were validated would also be ideal; consider work by Yardley (2016; https://doi.org/10.1080/17439760.2016.1262624) for guidance in this regard.- Do also consider the various careful and detailed comments provided by the reviewer below.

We look forward to receiving your revised manuscript.

Kind regards,

Joshua Uyheng

Academic Editor

PLOS ONE

Reviewers' comments:

Reviewer's Responses to Questions

**Comments to the Author**

1. Is the manuscript technically sound, and do the data support the conclusions?

Reviewer #1: Yes

2. Has the statistical analysis been performed appropriately and rigorously? 

Reviewer #1: N/A

3. Have the authors made all data underlying the findings in their manuscript fully available?

Reviewer #1: Yes

4. Is the manuscript presented in an intelligible fashion and written in standard English?

Reviewer #1: Yes

5. Review Comments to the Author

Reviewer #1: Dear Authors,

Thank you for the article and research you are conducting. The article offers important contributions to the field of racism, and racialisation in healthcare and medicine, not only within the German context but also for the broader research. As you know, most research does not explore the way healthcare providers, including students, discuss racism and how this may impact healthcare interactions and medical decisions. Hence, the article is an important step towards understanding how racialisation works in healthcare. However, there are some aspects of the article that need more work, especially in regard to defining some main concepts, and the structure of the results. I will divide the comments into main comments, and proceed to more detailed comments:

Main comments

• The article offers important insights. However, some concepts need to be defined in the introduction. Moreover, some new concepts are introduced in other sections and not discussed in the introduction. I suggest going through the article, writing down all the concepts and ensuring they are properly defined in the introduction. These concepts should form your conceptual framework which is a bit unclear at the moment. For instance, you state in line 207 in the method section that “The qualitative analysis was guided by the assumption that racism influences peoples’ thinking and acting both consciously and on an unconscious or habitual level,” but you do not offer the reader a discussion on this issue i.e., the embeddedness of racism in the various institutions in the nation-state. Clarity about your concepts and the conceptual departure points would make the article stronger and offer a reading an understanding of what guided your research and analysis. Below I will illustrate the undefined or unclear concepts:

Concepts not defined: 1. habitual approach and habitus; 2. racialized categorization -this concept is not introduced in the introduction- 3. antiracism – needs to be defined as you state that the category criticism holds a superior moral ground (rather, it is the participants who employ a critical view of racism). If criticism is a superior moral ground, what constitutes antiracism?) – 4. medical professionalization and professionalism, and its connection to habitus need to be unpacked. 5. Pointed labels is another undefined concept not discussed or introduced in the introduction. Finally, 6. heuristic tool mentioned in the result section needs to be defined, and introduced in the introduction.

I would suggest not only going through the concepts and ensuring that they are defined but also limiting the number of terms/concepts that are used, as some of them are not necessarily employed theoretically, such as heuristic tools and pointed labels.

• The aim of the article is not consistent, and the wording changes throughout the article. For instance, in the abstract, the aim is stated as: “To understand how medical students in Germany deal with the topic of racism,” while in the introduction, the aim stated is “we seek to contribute to a better understanding of the conditions of and obstacles to discussing racism in medical education. The focus is on the medical habitus and habitual approaches to the issue of racism in discussions among medical students.” In the method section, a new aim is introduced, which is: “to explore the practical approaches of medical students in addressing racism”. These different aims are not the same and require different ways to analyse and thus structure and delineate the results. My suggestion from reading the result section is that the aim is what is stated in the abstract.

• The method section includes a description of the analysis; in line 235, you state that you have analysed the performance and interactional aspects in the group discussions, which, in the result section, you delineate separately with the title On the level of performance and interaction. I am not convinced that such differentiation is necessary or adds anything to the article since focus groups are, per definition, interactional, and the interactional aspect is always part of any analysis. My suggestion is to remove the separate title and integrate the interactional parts with the analysis of each theme.

• Another issue with definitions is how the result section is structured. My main issue here is that the definitions of the themes provided at the beginning of the result section do not always coincide with what the results show. For instance, “Scientism” is defined as a way to approach racism “from a viewpoint of presumed scientific objectivity and neutrality, and interprets it as a problem of ignorant or malevolent misuse of scientific knowledge.” However, this is not what the results show. Rather than the definition provided, the results show that scientific knowledge is used to justify the existence of ‘race’ or racialised categories, i.e., that ‘race’ is indeed a biological reality rather than a product of racism. Hence, it seems that it is not that racism is viewed as non-scientific; rather, ‘race’ is seen as a scientific concept, which is a very interesting finding. My suggestion is to provide the correct definition of this theme so it is consistent with what the actual result shows, such as calling it scientism as justification for the existence of ‘race’ or something of that nature, but it is up to you to decide a new labelling.

• In regard to the theme, pragmatism; It is defined at the beginning of the result section as emphasising “the importance of practical professional experience, and views racialized categorizations as a well-tried, necessary, and useful heuristic tool to cope with the hustle and bustle of everyday clinical practice.” You also state in line 342 that “This approach is characterized by the habitus of a (future) physician who is mainly interested in hands on solutions for the everyday challenges of patient care.” Rather than emphasising the importance of practical professional experience, it seems that the onus is put on using ‘race’ to justify pseudo-medical categorisations and diagnoses such as Morbus mediterraneus, etc., i.e., race talk that is used to justify diagnosis on the basis of racialisation. I think the wording of the theme could be reformulated to convey its message. It is not pragmatism per se. Rather, it is using so-called medical pragmatism as justification for racial categorisation.

• Regarding the theme “Relativism”, I find it the most unclear one. You define it as “characterized by a combination of unreserved trust and obedience vis-a-vis authoritative medical textbook knowledge and standards on the one hand, and pronounced uncertainty and subjectivism in ‘soft’ matters like the social recognition and moral evaluation of racism on the other.” This definition needs to be unpacked as it is unclear, especially as you provide another definition in line 429: “Relativism is the claim of having not enough knowledge about racism to form a clear and definitive position.”. Please provide one consistent definition. Further, the term relativism does not convey anything in particular since one is left to wonder: relativism of what and in relation to what. Additionally, why is the claim of not having enough knowledge about racism termed relativism? These issues need to be clarified and addressed.

• Moreover, the themes do not convey the same logic concerning the analytical approach. This could be attributed to the various aims presented in the article. While scientism and pragmatism and, in a way, relativism seem to be about ways of understanding racism and ‘race’, interculturalism and criticism are about different ways to approach racism and ‘race’. So there is a jump in logic from the first three themes to the last two. This could be resolved by changing the way these themes are articulated, so it is not the approach that is emphasised but the way of understanding racism and ‘race’ that are emphasised.

• The discussion section should be further developed in regard to how participants understand the relation of ‘race’ to racism, which is currently absent, and in relation to other research in healthcare in this area.

Detailed comments:

1. Line 46: You state: “Talking about racism induces reflection, decreases racist beliefs and discrimination, and expands critical action against it [28-30].” Are you claiming that there is evidence that shows that discussion on racism results in decreased racial bias and beliefs? If so, explain how the provided references (The English ones since I unfortunately do not speak German) do not convey such an association.

2. Line 51. Please provide the year of publication after Novak et al. as well as the citation.

3. Line 58: Please change the word German to Germany.

4. Line 72: Spell out 22 (Twenty two) since numbers should be spelled out at the beginning of sentences.

5. The introduction lacks a review of research on racial bias. Please include this literature under the section Racism and anti-racism in medicine and healthcare in Germany.

6. You state in line 103 that: “ These observations about intention and the unconscious suggest that racism should be discussed with special regard to habitualized approaches, dispositions, and hence, habitus [22, 24-26, 68].” Please unpack this sentence. I suggest discussing these terms earlier since they seem to form your conceptual framework.

7. Line 123 forward includes a document analysis pertaining to the German CanMEDS framework, which should not be included in the introduction section since it is a result of your analysis. This could be the basis of another article. Moreover, this part is not necessary for the justification of this article.

8. Line 167 table 1: please revise the structure of the table. The numbers are not organised correctly.

9. Where minoritised people in the same group as majoritised people? how did this affect the discussion and your choice of methodology? Please reflect on this since you do state that the critical view on racism may have been limited due to the interactional aspect of group discussions? Reflect also on why individual interviews with minoritised people was not conducted.

10. Line 324: You mention the Holocaust but do not mention other genocides conducted by Germans such as in Namibia.

11. Line 359. The terms Morbus ….are they used in the medical curricula or are they race talk used in lectures etc.?

12. Line 521-528. How is this description of interculturalism different from the previous themes? Could you clarify this part? Similarly, in lines 556 and 557, you state that interculturalism is accompanied by a self-perception of being above racism. However, this seems to be the overarching theme and does not only pertain to this particular theme.

13. Please include research on racial bias in healthcare in the discussion section and delineate how your results speak to this body of research (majority of which is from the USA).

Overall, I enjoyed reading your article which offered very interesting results. Good luck with the review.

6. PLOS authors have the option to publish the peer review history of their article (what does this mean?). If published, this will include your full peer review and any attached files.

Reviewer #1: No

---

## [Author Response · Author response to Decision Letter 0]

25 Jul 2024

Reviewer #1

Reviewer#1 stated that our article offers important contributions to the field of racism and racialization in medicine and represents an important step towards understanding how racialization works in health care. At the same time, the reviewer pointed out that more work is needed, especially with regard to the definition of our central concepts and the structure of the results.

Main comments:

Reviewer #1 pointed out that our conceptual framework was a bit unclear since important concepts were not defined in the introduction and further concepts introduced in other sections of the text. In particular, this concerned the concepts of habitus and habitual approach, racialized categorization, antiracism, medical professionalization (and its connection to habitus). The reviewer suggested that we go through the text and identify a limited number of core concepts for analysis that are defined in the introduction and form our conceptual framework. 

-> We would like to thank the reviewer for this very helpful comment and the instructive suggestions! In order to clarify the conceptual framework of our analysis, we made three revisions: As suggested by the reviewer, we first reduced the number of concepts to the core conceptual framework, eliminating concepts that were not essential for our analysis, especially “pointed labels” (marked-up version: l. 380) or “heuristic tools“ (marked up version: l. 568, 722, 981, 1077). We replaced the terms “habitual approach” and “medical professionalisation” with the more precise terms of “ways of dealing with” and “professional socialisation”. Secondly, we now provide short working definitions of all the main concepts in the introduction: antiracism (lines 36-38, now and in the following lines refer to the unmarked version), professional socialisation (l. 43-45) habitus (line 46), orientation of action (l. 47-48), racism (l. 87-91), races (l. 92-94), racialisation (l. 93-95) and racialised (lines 95-98). Thirdly, we elaborate a bit further on these concepts and their interconnections in the methods section (l. 240-250), also providing a concise overview of our conceptual framework for analysis in the form of a table (l. 151). 

Reviewer #1 noted that the aim of the article was not stated in a consistent way, pointing to varying formulations throughout the text. The reviewer pointed out that different formulations of aims require different methods and different ways of organizing and structuring the results and recommended to stick with the formulation of the aim stated in the abstract (“to analyse how medical students in Germany deal with the topic of racism in discussions”).

-> We can see how these different formulations of the article’s aims can appear confusing and even misleading. We followed the reviewer’s recommendation to stick with the formulation in the abstract that we aim to analyse “how medical students in Germany deal with the topic of racism” (lines 4-5, l. 66-67, l. 166). In line with this formulation, we now state consistently throughout the manuscript that the focus is on analysing medical students’ ways of dealing with the topic of racism in discussions (Introduction line 74, methods l. 158, 257, 271, 285). To further increase coherence, we also revised the headings in the results section accordingly and renamed the five “habitual approaches” identified in the analysis and presented in the section as five different “ways of dealing with the topic of racism in discussions”, namely a “scientistic way” (ll. 298), a “pragmatic way” (ll. 389), a “subjectivist way” (l. 474), an “interculturalist way” (ll. 547), and a “critical way” (ll. 645). In light of the aforementioned changes, we eventually also decided to adapt the manuscript’s title accordingly. It now reads “How do medical students deal with the topic of racism? A qualitative analysis of group discussions in Germany” (l.1-2)

Reviewer #1 criticized that separating the analysis of performative and interactional aspects of medical students’ ways of dealing with the topic of racism in discussions was unnecessary and did not add anything to the article since these aspects are integral to focus groups as such and form part of their analysis anyway. The reviewer recommended to remove these separate titles and integrate the respective aspects in the analysis of each type.

-> We would like to thank the reviewer for this suggestion that makes the results section much more integrated and readable. We followed the advice and removed the respective subtitles delineating the analysis of performative and interactional aspects in the results section. The description of each way of dealing with the topic of racism in discussions now integrates performative and interactional aspects throughout, for example l. 335-337 and l. 375-376 for type A, l. 462-473 for type B, l. 537-456 for type C, l. l. 595-607 for type D, l. 681-687 for type E.

Reviewer #1 also pointed out that the definition of the types provided at the beginning of the results section was not always consistent with what the description of the results subsequently actually showed. The reviewer suggested formulating the definition of the types in a way that is coherent with the respective description of results. 

-> Thank you for pointing out this inconsistency! We have deleted the whole passage at the beginning of the results section that was aimed to give a short summary overview of the different ways medical students deal with the topic of racism in discussions (l. 367-380 in the marked version). Furthermore, we also made efforts to provide short descriptions of each type that capture its main defining aspects as they are subsequently unfolded in the respective part of the results section (see below).

Overall, reviewer #1 remarked that there was a jump of logic between the first three and the last two types described in the results section since the former referred to different understandings of ‘race’ and racism whereas the latter rather addressed different ways to approach ‘race’ and racism.

-> We would like to thank the reviewer very much for addressing this crucial point. We now make clearer right from the start that our analysis and the identified ways of dealing with the topic of racism actually encompass both aspects: different understandings of ‘race’ and racism as well as different attitudes towards ‘race’ and racism that both can be related to an underlying orientation of action. Thus, we now explain in more detail in the introduction (l. 42-48, l. 133) and the methods section (l. 240-250, 269-289) that our analysis of medical students’ habitual ways of dealing with the topic of racism in discussions was aimed at both explicit understandings as well as more or less implicit orientations. We also revised the descriptions of each type to make sure that each description actually explicitly addresses these different aspects. For example, we now make clearer that the ‘scientistic’ (type A, ll. 298), ‘pragmatic’ (type B, ll. 379) as well as ‘subjectivistic’ (type C, ll. 474) (formerly ‘relativistic’) way are not only about understandings of ‘race’ and racism, but also about more implicit orientations. Likewise, we point out that the ‘interculturalist’ (type D, ll. 547) and ‘critical’ (type E, ll. 645) way are not merely about attitudes or approaches towards ‘race’ and racism, but also include specific understandings (see below). Finally, this is also mirrored in our rephrased definitions of each type which now include all these aspects (see below).

In particular, reviewer #1 recommended to define ‘scientism’ in a way that makes clear that this type views ‘race’ as an objective biological reality and a scientific concept.

-> We have changed the definition of ‘scientism’ accordingly to include this aspect. The respective definition now explains that a scientistic way of dealing with the topic of racism “orientates action towards the idea of medicine as an objective science, justifies the use of racial categories as scientific and defines racism based on intention” (l. 299-301).

Furthermore, reviewer #1 also suggested rephrasing the definition of ‘pragmatism’ to reflect that this type essentially includes the justification of diagnoses based on racial categorizations such as ‘morbus mediterraneus’.

-> Here, too, we have rephrased the definition of this type accordingly. The respective definition now explains that the ‘pragmatic’ way of dealing with the topic of racism “orientates action towards tacit rules of clinical practice, justifies the use of racialised categories as practical and defines racism as an interpersonal problem.” (l. 380-382).

With regard to ‘relativism’, reviewer #1 found the definition that we provided rather unclear and demanded that we offer a more consistent definition. Also, the reviewer criticized that it was not clear why we called this type ‘relativism’, at all, and what the claim of not having enough knowledge about racism had to do with relativism.

-> We agree that this type is the most difficult to encapsulate in one short concise definition since it involves quite different aspects. Nevertheless, we rephrased the description given in order to make clear that this way of dealing with racism in discussions “stands for a lack of clear orientation of action and emphasizing uncertainty and subjective assessment when it comes to racialised categorisations as well as racism.” (l.475-477) Furthermore, we agree with reviewer #1 that the label ‘relativism’ was a bit misleading for this type. Therefore, we have changed the label to ‘subjectivist’, reflecting that the respective way of dealing with the topic of racism involves the claim that racism is a matter of “subjective impressions and opinions” (l. 479).

Reviewer #1 asked that we elaborate the discussion section in order to address participants’ understanding of the relation between ‘race’ and racism, also in relation to other research in healthcare in this area.

-> We added elaborated discussions about how the different types of dealing with the topic of racism understand the relation of ‘race’ and racism, c.f. l. 775-784 for the ‘scientistic’ type A, l. 785-792 for the ‘pracmatic’ type B, l. 800-804 for the ‘subjectivist’ type C, l. 817-822 for the ‘interculturalis’ type D, l. 830-833 for the ‘critical’ type D. 

Detailed comments:

1. Reviewer #1 asked us to explain how the references provided l. 46 (now l. 49-52 in the unmarked copy) support the claim that discussion about racism decrease racist beliefs and discrimination.

-> Evidence from medical education research shows that explicitly addressing racism and having discussions on this topic is an important step to tackle racism. We changed the sentence to “Talking about racism is an important precondition to induce reflection, decrease racist beliefs and discrimination, and expands critical action”. We chose this more reluctant formulation since discussions about racism do not automatically lead to a decrease in racial bias and beliefs but can be an important precondition and an element of teaching activities about racism and antiracism in medicine and health care (this is what the cited literature conveys). To stress this, we added more English references, e.g. on simulation-based antiracist training in medical education that shows the importance of feedback and discussions on this topic or on the concept of white fragility that impedes discussions about racism (l. 50-52, endnotes 33-38).

2. Reviewer #1 asked that we provide the year of publication after Novak et al.

-> We inserted the year of publication after Novak et al. (l. 56) and after all other in-text citations of this kind (e.g., l. 58, 62, 890)

3. Reviewer #1 pointed out that we misspelt the word ‘Germany’.

-> We changed the word accordingly (l. 66)

4. Reviewer #1 asked us to spell out 22 at the beginning of the sentence.

-> We now spell out the number (l. 138).

5. Reviewer #1 advised that we include a review of research on racial bias under the section “Racism and anti-racism in medicine and healthcare”

-> We included a longer review of research on racial bias in the respective section of the manuscript, covering major research strands and results in this field as well as different levels of medicine and healthcare, including individual bias as well as bias in knowledge bases and research (l. 110-136). 

6. Reviewer #1 asked us to unpack the sentence l. 103 about the necessity to discuss racism in the context of habitus and suggested to introduce these terms earlier.

-> We deleted the sentence and now instead introduce the relevant terms (professional socialisation, professional habitus) earlier (l. 42)

7. Reviewer #1 remarked that the analysis of the German CanMEDS framework should not be included in the introduction and could be the basis for another article.

-> We deleted the part about antiracism and critical approaches to racism in the CanMed roles. Instead, we provided a definition of Anti-racism right in the beginning of the article (l. 35-38) and only kept the information on the competencies formulated in the NKLM (l. 151-156).

8. Reviewer #1 demanded that we revise the structure of table 1.

-> We re-organized table 1 (l. 191). We used two PLOSone-articles related to the field of medical education as inspiration to revise the organization of the numbers in table 1 (Gottschalk et al. 2023; Tey et al. 2012). We added thematic headers for numbers concerning the same topic (gender, age, academic year) and added the type of numbers provided (n, range, mean). Moreover, by providing the classifying keywords from the participants’ answers to the open-ended question about “experience of discrimination” in the pre-questionnaire, we made more transparent how we counted the participants’ “experiences of discrimination”.

9. Reviewer #1 asked whether minoritized and majoritized people were included in the same groups and how this influenced our methodological approach and the group discussions. The reviewer also asked us to explain why we did not conduct individual interviews with minoritized groups.

-> Thank you for pointing out this crucial methodological aspect of our study. It is correct that people with and without personal experiences of racism were included in the same groups. We revised our methods section to explain why we opted for this research design instead of individual interviews with minoritised people: We were interested in observing how discussions in such diverse groups would develop. Discussions about racism do not only occur in specifically prepared teaching environments but often in clinical or educational context where this topic is not the main focus and group composition may be heterogenous (l. 168-173). Furthermore, in the discussion section, we elaborate our discussion of the influence of the diverse group composition on the discussions and our interpretation of the results. In particular, we explain how critical ways of dealing with the topic of racism sometimes remained without great validation and resonance in the group discussions. It is not that students explicitly discouraged other students to display a critical way to deal with the topic, instead, it was more implicitly through how statements related to other statements instead of taking up what was brought to the fore by students who employed a critical perspective on the topic (discussion: l. 868-871, results: 725-734)

10. Reviewer #1 pointed out that in line 324 (first submitted copy) we mention the Holocaust but no other genocides committed by Germans such as in Namibia.

-> In this part of our result section (type A), we mention the Holocaust as part of the implicit and collectively shared knowledge that medical students refer to when they discuss the use of race categories in medicine in a ‘scientific’ way (l. 356-375). It is true that Germans participated in more racist crimes and genocides than the Holocaust, e.g. the Herero and Nama genocide in Namibia or the Romani Holocaust (‘Porajmos’) but the students only referred to the Holocaust in this sequence of the group discussions. Because of this, we decided to refrain from adding the genocide in Namibia as an external information to this section of the text. 

11. Reviewer #1 wanted to know whether terms like ‘

---

## [Decision Letter · Decision Letter 1]

8 Aug 2024

PONE-D-23-43307R1How do medical students deal with the topic of racism? A qualitative analysis of group discussions in GermanyPLOS ONE

Dear Dr. Gerhards,

Thank you for submitting your manuscript to PLOS ONE. After careful consideration, we feel that it has merit but does not fully meet PLOS ONE’s publication criteria as it currently stands. Therefore, we invite you to submit a revised version of the manuscript that addresses the points raised during the review process.

We look forward to receiving your revised manuscript.

Kind regards,

Emma Campbell, Ph.D

Staff Editor

PLOS ONE

On behalf of 

Joshua Uyheng

Academic Editor

PLOS ONE

Journal Requirements:

Additional Editor Comments:

At copyediting time, please kindly ensure that the title is consistent. There are different versions in the cover letter and the submitted manuscript. In my view, the version that reads "How do medical students deal..." is preferable. I would also suggest that the italicization of white (when referring specifically to the authors' racial identification) be omitted.

Reviewers' comments:

Reviewer's Responses to Questions

**Comments to the Author**

1. If the authors have adequately addressed your comments raised in a previous round of review and you feel that this manuscript is now acceptable for publication, you may indicate that here to bypass the “Comments to the Author” section, enter your conflict of interest statement in the “Confidential to Editor” section, and submit your "Accept" recommendation.

Reviewer #1: All comments have been addressed

2. Is the manuscript technically sound, and do the data support the conclusions?

Reviewer #1: Yes

3. Has the statistical analysis been performed appropriately and rigorously? 

Reviewer #1: N/A

4. Have the authors made all data underlying the findings in their manuscript fully available?

Reviewer #1: Yes

5. Is the manuscript presented in an intelligible fashion and written in standard English?

Reviewer #1: Yes

6. Review Comments to the Author

Reviewer #1: (No Response)

7. PLOS authors have the option to publish the peer review history of their article (what does this mean?). If published, this will include your full peer review and any attached files.

Reviewer #1: No

---

## [Author Response · Author response to Decision Letter 1]

9 Sep 2024

Reviewer #1

Reviewer#1 stated that all comments have been addressed and made no further comments. 

-> We would like to thank again Reviewer 1 for the detailed and very instructive review in round one. We are glad to hear that we addressed all points raised. 

Academic Editor

The academic editor asked us to ensure that the title is consistent and pointed to the fact that there are different versions in the cover letter and the submitted manuscript. The editor sees the version that reads "How do medical students deal..." as preferable. 

-> Thank you for pointing out this inconsistency. We had changed the manuscript title in the first round of revisions but still used the original title in the response letter. We now consistently use the new title “How do medical students deal with the topic of racism? A qualitative analysis of group discussions in Germany” that is found in the revised manuscript and this response to the reviewers. Accordingly, we also edited in the supporting information S1_Text (discussion guideline) that is intended to be published with the article and uploaded a revised version. 

The academic editor also suggested that the italicization of white (when referring specifically to the authors' racial identification) be omitted.

-> Thank you for this suggestion. We followed the suggestion and deleted the italicization of “white” throughout the document. 

Journal requirements

The journal asked us to review our reference list to ensure that it is complete and correct and to mention any changes to the reference list in the rebuttal letter. Especially the citation of retracted articles should be indicated as such in the References list and also include a citation and full reference for the retraction notice.

-> We checked again our reference list to ensure that it is complete and correct. We do not cite a retracted article. The article cited as reference no. 135 has been corrected but not retracted. We made the following changes to the reference list:

- Reference no. 2: Added “th” to “7th ed.".

- Reference no. 6: Deleted “Policy & Procedure Manual” in the title., deleted “Canada” and edited the date published and cited. 

- Reference no. 14: Edited Date according to format style.

- Reference no. 15: Added interpunctuation.

- Reference no. 25: Added “p” to “pp”

- Reference no. 26: Added “p” to “pp”

- Reference no. 28: Added "p” to “pp”

- Reference no. 30: Deleted unnecessary “1.”, added “p” to “pp”

- Reference no 32: Added DOI

- Reference no. 33: Deleted “Wiesbaden” in publisher’s name, added “p” to “pp”

- Reference no. 37: Edited capitalisation of the title

- Reference no. 39: Added “p” to “pp”

- Reference no. 40: Added “p” to “pp”

- Reference no. 45: Corrected page numbers and added “p” to “pp. 205-215”.

- Reference no. 51: Added “p” to “pp”

- Reference no. 52: Adeed “p” to “pp”

- Reference no. 66: Deleted PubMed ID

- Reference no. 67: Deleted PubMed ID

- Reference no. 68: Corrected typo in title, deleted “A” in publisher’s name. 

- Reference no. 70: Corrected typo in “USA” 

- Reference no. 76: Deleted “[1952]” as first published date according to the citation guidelines. 

- Reference no. 78: Added DOI

- Reference no. 82: Added DOI

- Reference no. 93: Deleted PubMed ID

- Reference no. 97: Deleted Epub date.

- Reference no. 116: Edited capitalisation of the edited book’s title, deleted unnecessary parts of the edited book’s subject, added “p” to “pp”

- Reference no. 120: Added “p” to “pp”

- Reference no. 123: Added “p” to “pp”

- Reference no. 133: Added editor and edited capitalisation of the edited book’s title. Added place published and publisher. Added “p” to “pp”.

- Reference no. 137: Changed the journal’s title to the abbreviated form (ISO 4) 

- Reference no. 137: Added “p” to “pp”

- Reference no. 138: Added “p” to “pp”

- Reference no. 140: Edited incorrect information about the journal’s title and changed the correct journal’s title to the abbreviated form (ISO 4): “Am Psychol”

---

## [Decision Letter · Decision Letter 2]

16 Oct 2024

PONE-D-23-43307R2How do medical students deal with the topic of racism? A qualitative analysis of group discussions in GermanyPLOS ONE

Dear Dr. Gerhards,

Thank you for submitting your manuscript to PLOS ONE. After careful consideration, we feel that it has merit but does not fully meet PLOS ONE’s publication criteria as it currently stands. Therefore, we invite you to submit a revised version of the manuscript that addresses the points raised during the review process.

We look forward to receiving your revised manuscript.

Kind regards,

Rahul Sambaraju

Academic Editor

PLOS ONE

**Journal Requirements:**

**Additional Editor Comments:**

Dear author,

I have now received all the reviews for your submission. As you will see, one reviewer has recommended that we accept your paper and another reviewer has some suggestions to improve the quality of your submission. My own position aligns with the latter. Please take-up these minor revisions that largely deal with the method and results section. We can then proceed with further processing.

Thank you for your work and this wonderful paper. We look forward to receiving your revised paper.

Best

Rahul

Reviewers' comments:

Reviewer's Responses to Questions

**Comments to the Author**

1. If the authors have adequately addressed your comments raised in a previous round of review and you feel that this manuscript is now acceptable for publication, you may indicate that here to bypass the “Comments to the Author” section, enter your conflict of interest statement in the “Confidential to Editor” section, and submit your "Accept" recommendation.

Reviewer #1: All comments have been addressed

Reviewer #2: (No Response)

2. Is the manuscript technically sound, and do the data support the conclusions?

Reviewer #1: Yes

Reviewer #2: Yes

3. Has the statistical analysis been performed appropriately and rigorously? 

Reviewer #1: N/A

Reviewer #2: N/A

4. Have the authors made all data underlying the findings in their manuscript fully available?

Reviewer #1: Yes

Reviewer #2: No

5. Is the manuscript presented in an intelligible fashion and written in standard English?

Reviewer #1: Yes

Reviewer #2: Yes

6. Review Comments to the Author

**Reviewer #1:** Dear Authors,

As explained in the previous round of revisions, all comments have been addressed. The article offers important contributions that will aid in understanding the production of racialisation in healthcare.

**Reviewer #2: **Review on “How do medical students deal with the topic of racism?”

Dear colleagues, I read your study with great interest. Below is the summary of the article with my impressions. I then give specific suggestions that I would recommend you address in your revision.

This article analyzes German-speaking medical students speaking about the topic of racism in online group conversations. The authors provide a good introduction and literature review that introduces how racism, race, and bias is thought about in this article. The qualitative study using virtual focus group interview data is relevant to the research questions as they give both a thematic overview of the ideas that medical students have related to questions of racism, and also show how medical students explain their understanding of racism and race. The results are presented in a clear manner and use relevant quotes, and the tables support the results well. The discussion and conclusion of the paper has some useful takeaways.

Based on this positive assessment, I have some comments that I think are important for the authors to take into account, and I have some suggestions that you can consider:

Literature review: The discussion of racism as an ideology and system is well argued. Paying attention to the implicit forms of racist meaning making is an important point. However, I’d make the transition from Fanon’s point to racial bias more clear (110 to 111). I think that these two interpretations are slightly different, and that moving from Fanon’s explanation that racism structures “implicit knowledge” (line 108) is not the same thing as implicit bias which connects more to a cognitive argument of explaining racism. Please make this transition more clear, and articulate why you see both of these explanations as necessary. Also, you could return to this question of implicit bias in the conclusion - and discuss if this still hold its relevance or if other explanations, like systemic ones might be more relevant.

Methodology: Write who translated the quotes from your material.

Findings: The findings are clear and the quotes relevant. However, I’ve got some questions/concerns regarding the types. I don’t think that a typology of how medical students deal with racism is a helpful description of what’s happening. Rather than types, as you say yourself, these are 5 typical ways that German-speaking medical students deal with the topic of racism; and each person could potentially adapt each way. I suggest that you don’t use the word types to describe it but something else, for example explanations. The term type evokes the idea of typology, which can be associated with fixity, an inability to change. I think an important point that you’re making by using the approach of habitus is that we are socialized into certain behaviors and thoughts, and that we can also change. I think that this should be reflected in the naming of the 5 typical ways. One other word I don’t find so fitting is “scientistic.” I understand that you want to use adjectives to describe each typical way, but I think saying “scientist” would be sufficient.

Discussion: This is a good discussion in which you discuss the main takeaways. I wonder if the language could be changed slightly in lines 845-847. You write that “Black women experience to be hypersexualised…” By writing experience to be, I get the impression that this could be debatable. I suggest that you instead write “Black women are hypersexualized” to demonstrate that this in fact happens, and it is not just from their experiences. Finally, it might add to your article to discuss specifically the need for more interdisciplinary work within medical education. Specifically, the lack of critical literacy in terms of understanding racism as a system demonstrates the need for more education outside of the “hard” sciences.

I hope that these requests and suggestions are helpful for the authors to improve the quality of this study.

7. PLOS authors have the option to publish the peer review history of their article (what does this mean?). If published, this will include your full peer review and any attached files.

Reviewer #1: **Yes: **Sarah Hamed

Reviewer #2: No

---

## [Author Response · Author response to Decision Letter 2]

21 Oct 2024

Reviewer #1 (Sarah Hamed):

Reviewer#1 confirmed that all comments had been addressed in the previous round of revisions and underlined that the article will help readers understand the production of racialisation in healthcare.

->We would like to thank Reviewer #1 again for the detailed and very instructive review in round one. We are glad she evaluated our article positively.

_ _ _ _ 

Reviewer #2:

Reviewer #2 summarized the articles’ content, gave overall positive feedback, and suggested minor revisions to address five aspects.

-> We thank Reviewer #2 for the encouraging feedback and helpful comments that point to important aspects. Addressing them helped us improve and clarify the article’s argument.

1. Regarding the background section, Reviewer #2 recommended clarifying the argumentative transition from Fanon’s perspective on “implicit knowledge” on the one hand and the more cognitive take on physicians’ racist behaviour with the concept of “implicit bias”. Moreover, Reviewer #2 proposed returning to the relevance of implicit bias research compared to systemic explanations in the discussion.

-> Thank you for the close reading and for identifying this argumentative weakness in the article’s background section. We tackled this inconsistency by linking Fanon’s analysis of “implicit knowledge” more explicitly to the concept of habitus that connects the individual and collective/structural aspects of racism (105-113). Moreover, to distinguish this perspective from the one of implicit bias research, we now begin the paragraph about implicit bias with the explanation that research on implicit bias “takes a closer look at the cognitive processes […]” (114-115) involved in racism. Furthermore, we added another sentence in the discussion section highlighting the importance of analysing structural aspects for understanding the persistence of racism in medicine and health care (861-863).

2. Regarding the article’s methods section, Reviewer #2 suggested mentioning who translated the quotes from the material.

-> Thank you for this suggestion! We added the information in line 234: “the quotes from the material were translated into English by the authors.”

3. Reviewer #2 shared the impression that calling the five identified ways of dealing with the topic of racism a ‘typology’ evokes the impression of them being “fixed” with no possibility to change or adapt. Reviewer #2 recommends choosing a different wording. They also suggest calling the first identified way of dealing with the topic of racism “scientist” rather than “scientistic.”

-> We have revised the whole text to avoid terminology suggesting a systematic typology with fixed types. Instead, we now speak of typical ways, approaches, or perspectives in dealing with racism. However, after careful consideration, we decided to stick to the term “scientistic” for three reasons: First, as Reviewer #2 noted, we use adjectives for all five ways of dealing with the topic of racism and would like to keep this nomenclature consistent. Secondly, we think this use of adjectives actually helps to signal that we analyse typical “ways” (modes of talking and acting) rather than fixed types of people. Thirdly, we are concerned that the term “scientist” might be misleading since it might suggest a scientific approach to racism. We think “scientistic” (derived from the position of scientism) makes clearer that we talk about a professional habitus that refers to (a particular understanding of) science to justify positions about racism and lend them authority.

4. Regarding the discussion, Reviewer #2 suggested changing how we report the findings from another study about the hypersexualization of black women. Instead of writing about the experience of black women to be hypersexualised, Reviewer #2 proposes to change the language to make clear that it is not only an experience of black women but something that in fact happens.

-> We changed the sentence according to Reviewer #2’s suggestion. It now reads: “Black women are hypersexualized and more likely to be checked for STIs” (851).

5. Regarding the discussion, Reviewer #2 also suggested discussing the need for more interdisciplinary collaboration and research with special regard to fostering the understanding of racism as a system by including, e.g., sociological perspectives into medical education.

-> Thank you for this important point! We came to the same conclusion and, therefore, added an explicit statement about the need to include more institutionalized interdisciplinarity in medical education (1008-1011).

---

## [Editor Report · Decision Letter 3]

29 Oct 2024

How do medical students deal with the topic of racism? A qualitative analysis of group discussions in Germany

PONE-D-23-43307R3

Dear Dr. Gerhards,

We’re pleased to inform you that your manuscript has been judged scientifically suitable for publication and will be formally accepted for publication once it meets all outstanding technical requirements.

Kind regards,

Rahul Sambaraju

Academic Editor

PLOS ONE
---

## [Editor Report · Acceptance letter]

9 Nov 2024

PONE-D-23-43307R3 

PLOS ONE

Dear Dr. Gerhards, 

I'm pleased to inform you that your manuscript has been deemed suitable for publication in PLOS ONE. Congratulations! Your manuscript is now being handed over to our production team.

Kind regards, 

on behalf of

Dr. Rahul Sambaraju 

Academic Editor

PLOS ONE